# Man-made microbial resistances in built environments

Alexander Mahnert [1], Christine Moissl-Eichinger[2,3], Markus Zojer[4], David Bogumil[5], Itzhak Mizrahi[5], Thomas Rattei [4], José Luis Martinez[6] & Gabriele Berg[1,3]

Antimicrobial resistance is a serious threat to global public health, but little is known about the effects of microbial control on the microbiota and its associated resistome. Here we compare the microbiota present on surfaces of clinical settings with other built environments. Using state-of-the-art metagenomics approaches and genome and plasmid reconstruction, we show that increased confinement and cleaning is associated with a loss of microbial diversity and a shift from Gram-positive bacteria, such as *Actinobacteria* and *Firmicutes*, to Gram-negative such as *Proteobacteria*. Moreover, the microbiome of highly maintained built environments has a different resistome when compared to other built environments, as well as a higher diversity in resistance genes. Our results highlight that the loss of microbial diversity correlates with an increase in resistance, and the need for implementing strategies to restore bacterial diversity in certain built environments.

[1] Institute of Environmental Biotechnology, Graz University of Technology, Petersgasse 12/I, Graz 8010, Austria. [2] Department of Internal Medicine, Medical University Graz, Auenbruggerplatz 2, Graz 8036, Austria. [3] BioTechMed Graz, Mozartgasse 12/II, Graz 8010, Austria. [4] Division of Computational Systems Biology, Department of Microbiology and Ecosystem Science, University of Vienna, Althanstrasse 14, Vienna 1090, Austria. [5] Department of Life Sciences, Faculty of Natural Sciences, Ben-Gurion University of the Negev, Box 653, Beer-Sheva 84105, Israel. [6] Centro Nacional de Biotecnologia, CSIC, Calle Darwin 3, Madrid 28049, Spain. Correspondence and requests for materials should be addressed to A.M. (email: alexander.mahnert@gmail.com)

The increased morbidity and mortality rate associated with infections by antibiotic-resistant bacteria is one of the main global threats human kind has to face nowadays. Antimicrobial resistance (AMR) is recognized as a real health crisis that has to be forcefully tackled on several fronts[1]. Most of these fronts are directly linked to human behavior in the environment[2]. In recent years, research has been focused among others on the environmental dimension of AMR especially in livestock farming, waste water treatment, and in hospital settings. However, other built environments in which people commonly spend most of their lives (e.g., private homes and workplaces) have been often neglected in these studies, despite their potential relevance for the emergence and spread of AMR. An exception is the study by Lax and coworkers, who investigated AMR not only in a hospital setting[3], but also in private homes[4].

In line with such demands by the scientific community[2,5], we compared associations of AMR between surfaces in clinical settings and other built environments. We focused on three main aspects of anthropogenic influences on buildings: (1) occupancy and type of access, (2) room's usage, and (3) human activities that may alter the microbiota like microbial control and cleaning, in general. Since prior studies have already indicated that microbial community and resistome structures correlate with human actions in their environment[3,4,6,7], we were interested in learning how microbial control and building confinement affect the composition and functional capabilities of the residing microbiome with an in-depth analysis of the resistome and its mobile genetic elements. For this purpose, we defined a set of model built environments, which differ in their grade of anthropogenic influences, including microbial control, cleaning, and access. On one hand, we investigated different naturally unrestricted buildings (UBs) and houses with a high level of influence from the surrounding outdoor environment, including plants, in a rural setting. On the other hand, we sampled controlled built environments (CBs) with an increasing level of microbial confinement and cleaning operations from intensive care units (ICUs) to spacecraft assembly cleanroom facilities in urban areas. All samples were supported by a rich collection of environmental metadata to correlate compositions and functions of the microbiome with environmental parameters. This unique study design was further supported by a new sampling methodology to acquire deeply sequenced shotgun libraries even from low-biomass environments. In addition, a state-of-the-art genome centric bioinformatics analysis[8] was conducted to elucidate resistance features in their genome context.

These new insights are useful to model human-driven processes affecting in-house microbiota and its associated resistome and to improve our assessments on the possibilities of preserving or, eventually, designing microbiomes in built environments.

## Results

**Confinement correlates with reduced microbial diversity.** Opposing structures of UB and CB were accompanied by a significant loss (Spearman's rank correlation rho, correlation coefficient: -0.8783, $P = 0.02131$; one-sided $t$ test: $n = 9$, $t = -3.2$, df = 2.6, $P = 0.03$) of taxonomic diversity (Shannon–Weaver indices: CB 7.2 H', UB 8.8 H') (Fig. 1a and Supplementary Table 1). In contrast, the functional diversity between UB and CB remained balanced (10.8–11.1 H' according to SEED annotations; Fig. 1b). The analysis of 16S rRNA sequences showed even clearer differences between CB (5.6 H') and UB (7.2 H') due to lower diversity estimates for ICU samples (3.8 H') and a higher diversity for private houses (6.4 H') (Supplementary Fig. 1). These differences in diversity estimates were observed in the presence of constant bacterial abundances (~ $10^6$–$10^7$ 16S rRNA gene copies per m²), with a higher variability for the fraction of intact cells (~$10^3$–$10^7$ 16S rRNA gene copies per m²). However, diversity estimates did not correlate with the proportion of intact cells (Spearman's rank correlation rho, correlation coefficient: 0.2, $P = 0.4$).

**Environmental differences correlate with the microbiome.** Shotgun metagenome samples from public buildings and public houses were more similar to each other than samples obtained from private houses according to Principal Coordinates Analysis (PCoA) ordinations and Unweighted Pair Group Method with Arithmetic Mean trees. Even greater dissimilarities were observed between samples from UB and CB. Moreover, 16S rRNA-based population structure indicated lower dissimilarities for UB (mean Bray–Curtis distance 0.71) than for CB environments (mean Bray–Curtis distance 0.82; Fig. 2 and Supplementary Fig. 2).

Different categories of sampled built environments could be characterized by distinct compositions of the metagenomic reads even on the superkingdom level (Supplementary Fig. 3). Hence, proportions of bacteria vs eukaryota (mainly sequences assigned to humans) decreased significantly (one-sided $t$ test: $n = 9$, $t = 3.4$, df = 2.0, $P = 0.04$) from UB (~ 99% bacteria, ~ 1% eukaryota) towards CB (for bacteria: cleanroom ~ 69% and its gowning area ~ 85%; ICU ~ 55%). A similar pattern could be observed for archaea, although not significant (one-sided $t$ test: $n = 9$, $t = 1.9$, df = 2.0, $P = 0.1$), with higher counts (~ fourfold) in CB. Traces of viruses were less apparent between CB and UB, but showed highest relative abundances in the ICU and in the environment of public houses. Clear differences continued into higher taxonomic levels (Supplementary Fig. 4 and Supplementary Fig. 5): on the phylum level, public buildings and public houses were dominated by sequences of *Actinobacteria* (up to 50%) and *Proteobacteria* (~ 21%). In private houses, the proportion of *Firmicutes* raised up to 55%. Likewise, the proportion of *Firmicutes* was also higher after masking the DNA of compromised cells with propidium monoazide (PMA). In CB, the prevalence of bacterial phyla was reduced and proportions of multicellular organisms and not assignable sequences increased (up to 62% in the cleanroom). Furthermore, *Pseudomonas*, *Porphyromonas*, *Propionibacterium*, and *Prochlorococcus* could be identified as significant discriminative features (Supplementary Fig. 6) in CB by LEfSe (linear discriminant analysis of the effect size) analysis. Besides these bacterial taxa, also viral sequences (e.g., human herpes and papillomavirus) and assignments to arthropods (e.g., mites like *Trombidiformes* and *Prostigmata*) and insects (e.g., lices such as *Liposcelis bostrychophila* and cockroaches like *Blattella germanica*) were defined as discriminative features for CB.

The core 16S rRNA gene microbial profile was visualized in a core operational taxonomic unit (OTU) network (Supplementary Fig. 7). This analysis indicated a high proportion of shared OTUs assigned to *Acinetobacter* and *Staphylococcus* as well as a bigger overlap of samples from the cleanroom facility and unrestricted buildings compared to the core of samples from the ICU environment.

To correlate microbial community composition with environmental parameters, a bioenv test with Spearman rank correlations compared to Euclidean distances was applied on the 16S rRNA gene profile. This bioenv analysis showed higher correlations of samples with latitude, longitude, and sea level (best variable combination $\rho$w = 0.9425) than with temperature, humidity, and room variables, like the surface area, room height, or room volume (best variable combination $\rho$w = 0.7518). These correlations were further visualized as vectors on an Non-metric multidimensional scaling (NMDS) ordination of the sampled communities together with calculated ellipses per sampling

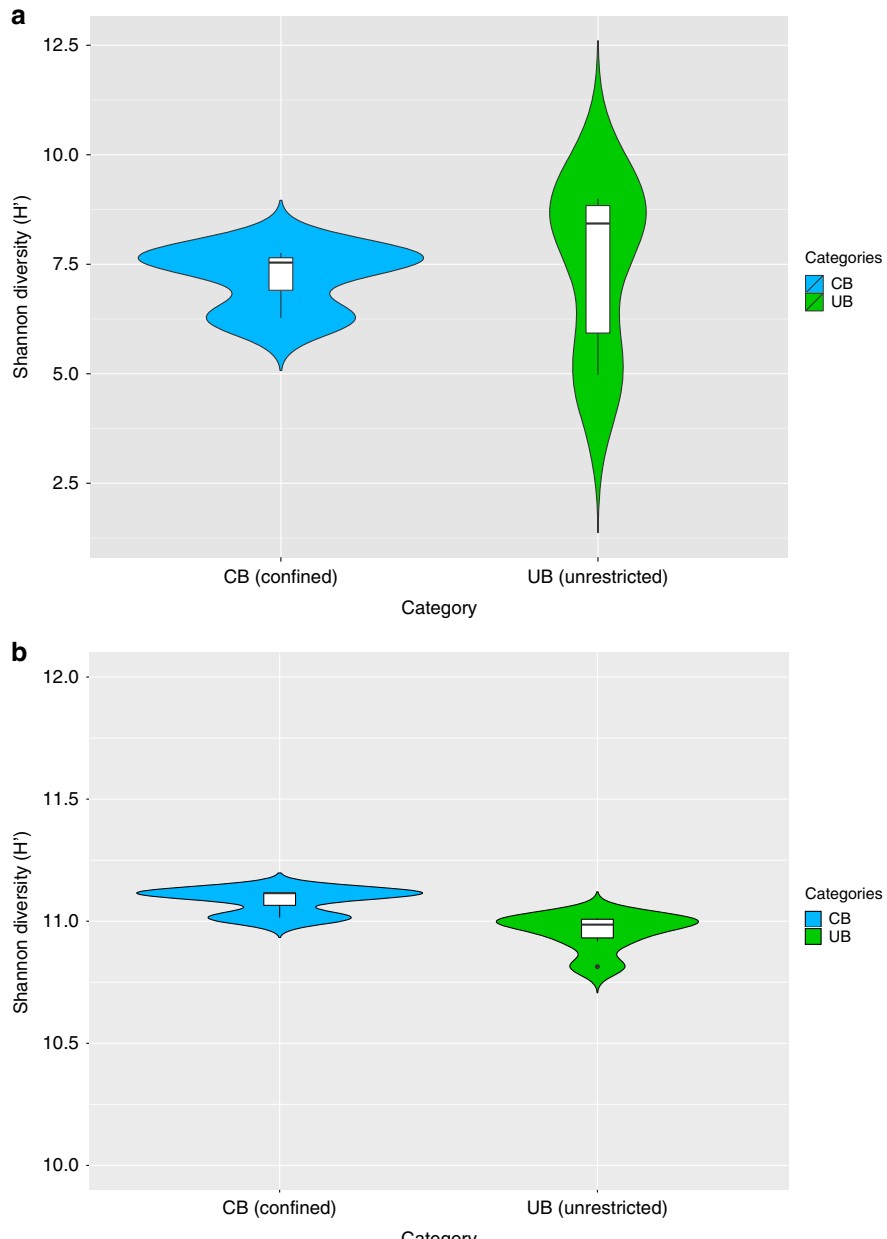

**Fig. 1** Microbial diversity estimates. Calculations were executed in MEGAN according to the results of the BLASTx searches against NCBInr. Data of single reads were filtered (unassigned reads were removed) and normalized (randomly and repeatedly subsampled to the smallest sample size). Violin plots showing the kernel probability density of the data, including a box with the median and the interquartile range, were created in R. **a** Significant differences of Shannon diversity estimates of microbial communities on species level in CB (confined) and UB (unrestricted built environments). **b** Similar Shannon diversity estimates of microbial functions on highest SEED levels (individual functional gene levels, level 5) in CB (confined) and UB (unrestricted built environments)

category (Fig. 3). This ordination showed distinct clusters for samples obtained from the surface of tiles in private houses, the sanitary environments in public houses and public buildings, or that ICU floors and ICU workplaces overlapped with samples from medical devices. However, associations of the microbiome with environmental variables like biogeography or microclimate could not be further supported or differentiated due to confounding variables (see Supplementary information).

In general, the composition of the microbiome was so distinct that the associated metadata categories could be predicted by supervised learning methods (random forest classification and regression models). Samples from CB or UB could be predicted with a high overall accuracy of 92%. Likewise, numerical

environmental parameters such as temperature ($R = 0.92$, $P = 4.8 \times 10^{-5}$), relative humidity ($R = 0.89$, $P = 3.3 \times 10^{-4}$), longitude ($R = 0.95$, $P = 2.8 \times 10^{-6}$), and sea level ($R = 0.82$, $P = 3.3 \times 10^{-3}$) could be easily predicted. Microbial abundances ($R = 0.63$, $P = 0.12$) and respective room areas ($R = 0.58$, $P = 0.24$) were not suitable to build predictive models from observed features.

**Changed functional capabilities were evident on genome levels**. Assembled contigs and scaffolds could be binned into 125 draft genomes (8–20 bins per sample). Most binned genomes were recovered from samples of private houses, while only a few genomes could be reconstructed from the ICU dataset

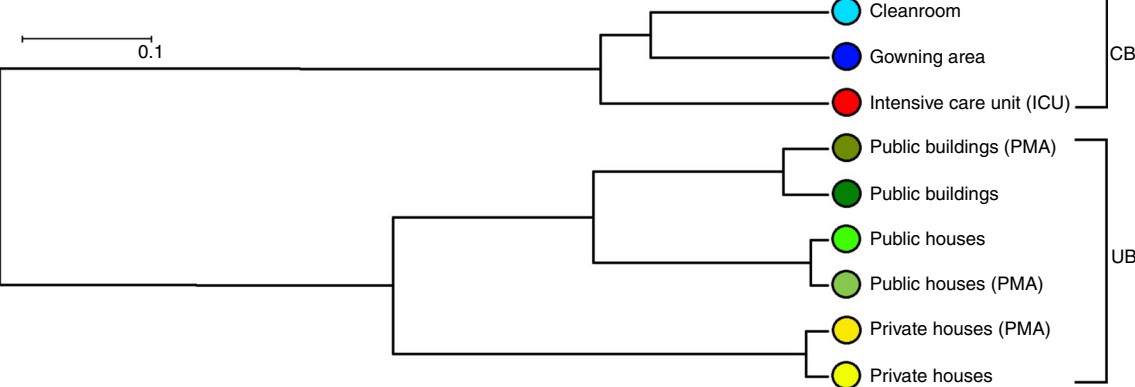

**Fig. 2** Connection between different built environment types. UPGMA tree (Unweighted Pair Group Method with Arithmetic Mean tree) of sampled built environments based on different microbial communities resolved to species level. Calculations were executed with MEGAN according to the results of the BLASTx searches against NCBInr. Data of single reads were filtered (unassigned reads were removed) and normalized (randomly and repeatedly subsampled to the smallest sample size). Color code for column environment: blue (cleanroom facility); red (intensive care unit); dark green (public buildings); light green (public houses); yellow (private houses)

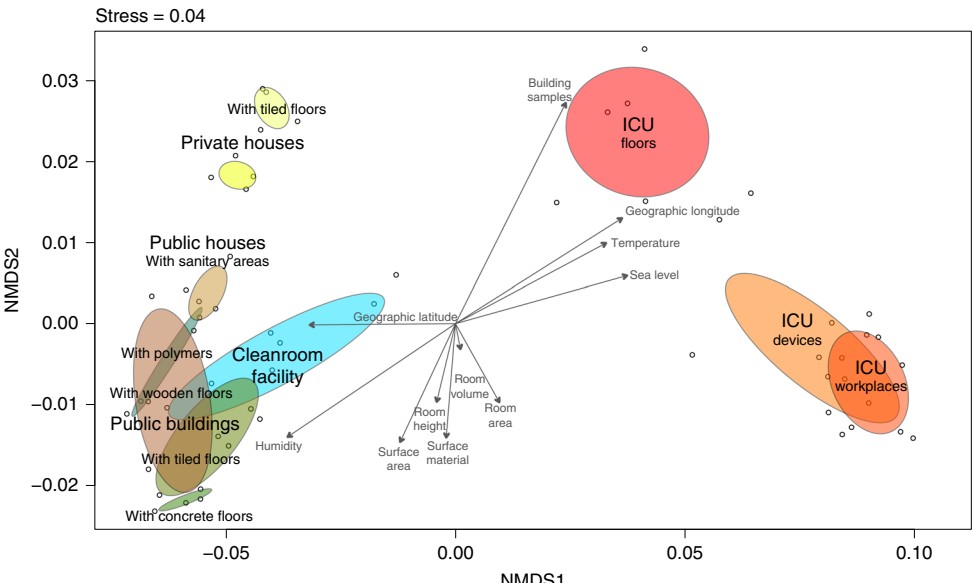

**Fig. 3** Environmental variables associated with the microbiome of sampled built environments. NMDS of 16S rRNA gene amplicons based on Bray–Curtis distances with superimposed vectors representing Spearman correlations of measured environmental variables (bioenv) based on Eucledian distances. Color code for column environment: blue (cleanroom facility); red (intensive care unit); dark green (public buildings); light green (public houses); yellow (private houses)

(Supplementary Table 2). A subset of 44 draft genomes (representing 45% of all assembled contigs) were sufficient in quality for an in-depth analysis. Annotations, replication activity, and predicted phenotypes of these binned genomes were significantly representative for CB or UB environments (Fig. 4). Hence, according to iRep, replication rates were lower in CB (two-sided two-sample Kolmogorov–Smirnov test: $D = 0.68$, $P = 0.005$) and ranged from 2 to 6 replication events for 10–75% of the sampled population. According to Phenotype Investigation with Classification Algorithms (PICA), several distinct phenotypes could be predicted (46 individual chi square tests, Bonferroni correction $P = 0.02$) on genome and marker-gene levels. Therefore, significant phenotypic traits for CB covered alkane degradation, benzoate degradation by hydroxylation, trimethylamine production by choline, T4 and T6 secretion systems, and plant pathogenicity based on thaxtomins, while arsenic detoxification and

facultative anaerobes were specific for UB. Overall, Gram-positive bacteria ($P = 0.004$) with functions associated with carbohydrate and amino acid metabolism dominated in UB. On the contrary, Gram-negative bacteria with many functions associated with virulence, disease ($P = 0.008$), defense ($P = 5.2 \times 10^{-5}$), and resistance ($P = 0.08$) were representative for CB ($P$ values were calculated by Kruskal–Wallis tests; Supplementary Figs. 8–11).

Genomes assigned to *Exiguobacterium* ($V = 0$, $P = 2.2 \times 10^{-11}$) and *Macrococcus* ($V = 0$, $P = 1.0$) were commonly recovered from diverse UB environments. Genomes of *Arthrobacter* ($V = 465.5$, $P = 2.9 \times 10^{-15}$) and *Janibacter* ($V = 0$, $P = 0.3$) were more specific for the category of public buildings and public houses. *Enhydrobacter* ($V = 0$, $P = 1.0$), *Kocuria* ($V = 0$, $P = 8.3 \times 10^{-4}$), and *Pantoea* ($V = 225$, $P = 1.2 \times 10^{-9}$) were found additionally in private houses together with *Lactococcus* ($V = 9$, $P = 1.0$) and *Staphylococcus* ($V = 3445$, $P = 0.01$). *Leuconostoc* ($V = 169$,

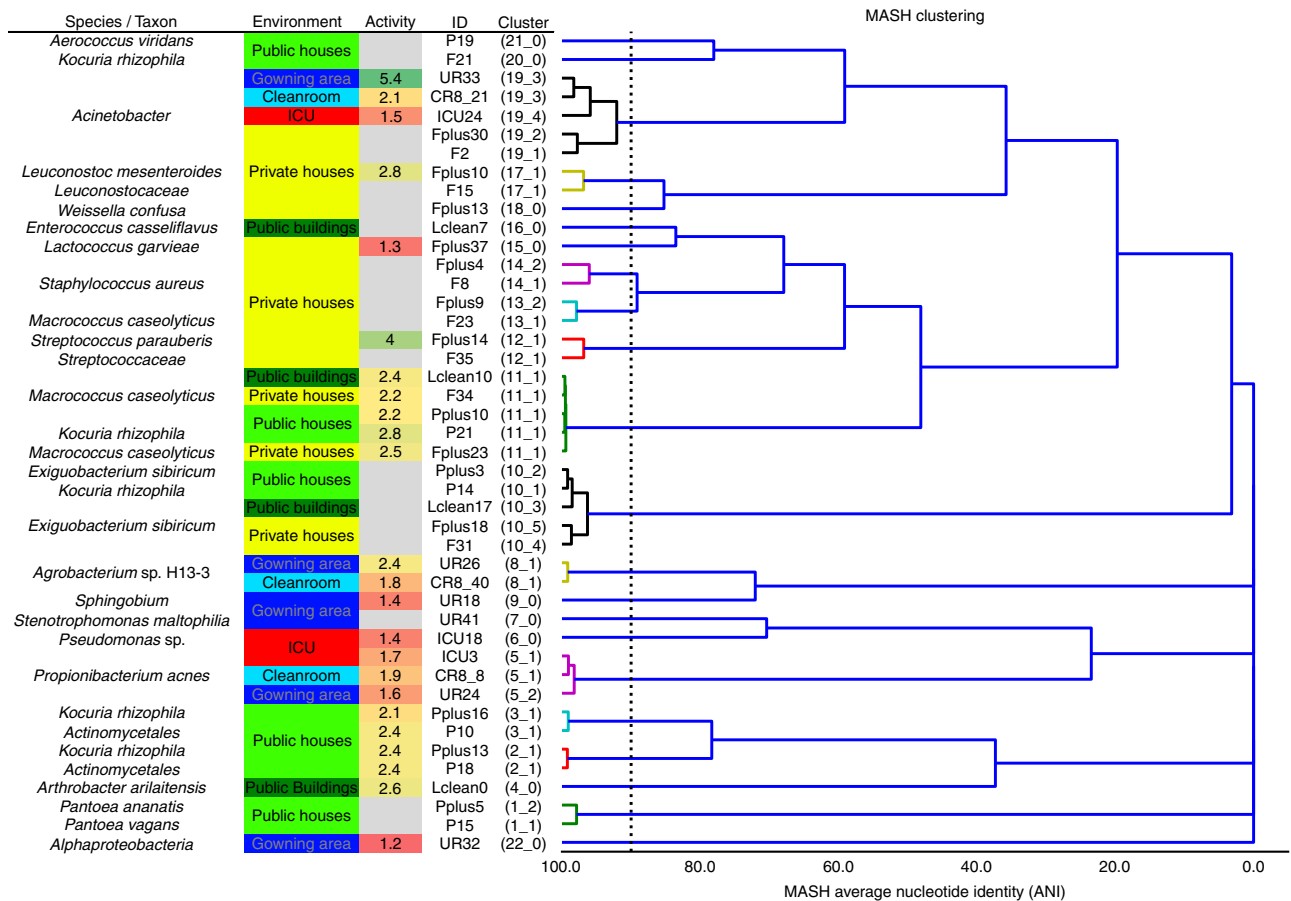

**Fig. 4** An overview of reconstructed genomes. High-quality binned genomes clustered by average nucleotide identity (ANI), resolved to highest taxonomic levels, respective built environment origins, and respective replication rates (activity). Color code for column environment: blue (cleanroom facility); red (intensive care unit); dark green (public buildings); light green (public houses); yellow (private houses)

$P = 0.9$) marked the transition from private houses to ICU. And finally, genomes assigned to *Propionibacterium* ($V = 2697$, $P = 0.01$), *Pseudomonas* ($V = 133530$, $P = 2.9 \times 10^{-15}$), and *Stenotrophomonas* ($V = 97.5$, $P = 0.07$) were characteristic to all CB environments ($P$ values from Wilcoxon signed rank tests; Fig. 4). Representative taxonomic assignments for distinct built environments were supported by data of the single-read analysis (Supplementary Figs. 5 and 6) and 16S rRNA gene amplicons (Supplementary Fig. 12).

Genomes assigned to the genus of *Acinetobacter* (median completeness 94%, median contamination 20%) were highly prevalent and ubiquitous in all sampled built environments. This has allowed a detailed comparison of closely related bacterial species from different maintained built environments regarding changed functional properties on pan-genome levels. Genomes of *Acinetobacter* from private houses, the ICU, the cleanroom and its gowning area shared a core genome with 24–39% of all CDS (proportion of core coding DNA sequences to all coding DNA sequences in a genome). Coding genes in the recovered genome of *Acinetobacter* (e.g., Acetyl-CoA acetyltransferase *fadA* or alcohol dehydrogenase *frmA*) from the ICU showed the biggest overlap with this core (39%) and less strain-specific CDS (784) than genomes of *Acinetobacter* from the private houses (2857 strain-specific CDS, 24% of the core genome). Regarding all binned genomes, the ICU environment showed the greatest density (highest grade of similarity) for its core genome (0.2% core CDS) compared to all other sampled built environments (Supplementary Table 3). Differences in the pan-genome of

*Acinetobacter* were especially striking for functions associated with virulence, disease, and defense. In CB, the number of assigned functions to these categories almost doubled compared to UB.

In general, functional traits were more evenly distributed over all sampled indoor spaces compared to microbial profiles (Supplementary Figs. 13–16). Nevertheless, a detailed LEfSe analysis based on SEED annotations revealed functions associated to Gram-positive bacteria (Gram-positive cell wall components, heme and hemin uptake, and utilization in Gram positives), fatty acid metabolism (fatty acid lipids, isoprenoids, teichoic and lipoteichoic acid biosynthesis), DNA repair systems (DNA repair *UvrABC* system, DNA repair bacterial Rec FOR pathway, and transcription repair-coupling factor), and heatshock (heatshock *dnaK* gene cluster) as significant discriminative features of UB. On the contrary, functions associated with Gram-negative bacteria (Gram-negative cell wall components), iron acquisition (ferrichrome iron receptor, *TonB*-dependent siderophore receptor, and siderophore pyoverdine), oxidative stress, membrane transport and secretion (*Ton* and *Tol* transport systems, RND efflux system inner membrane transporter *CmeB*, Type III, IV, VI ESAT secretion systems), virulence (virulence disease and defense), and resistances (resistance to antibiotics and toxic compounds, multidrug resistance efflux pumps, cobalt zinc cadmium resistance protein *CzcA*) were identified to be representative for CB. A comparison of all annotated SEED functions with the RAST server[9,10] revealed a high proportion of functions associated with amino acid and carbohydrate

metabolism for UB (Supplementary Fig. 17). In contrast, genomes from CB indicated a shift towards other functions like virulence, disease, and defense. Especially, genomes from the cleanroom environment showed much more evenly distributed functional capabilities for all functional groups and, additionally, many functions associated with stress response.

**Differences were reflected by the resistome**. Due to distinct profiles and our interest in functions related to virulence and resistance, we captured the virulome (entity of virulence factors) and resistome (entity of resistances against antibiotics) of CB and UB in greater detail. Slightly more virulence genes (VFDB) were detected for genomes of CB (19) than of UB (18). Highest proportions of virulence genes were present inside the ICU, followed by public and private houses. Lowest counts were visible for the highly unrestricted environment of public buildings. Hence, chromosomally encoded bacterial virulence in CB and UB was likely associated with its distinct microbial profiles. However, differences in proportions were not significant.

Compared to the virulome, the resistome showed clearer differences for CB vs UB. Using CARD (Comprehensive Antibiotic Resistance Database), 377 different resistance features could be identified for the 42 selected high-quality binned genomes and 91 extracted plasmids. Detected resistance genes were manually curated (removal of only mutation and regulation-mediated resistances according to ref. [11]) for a detailed analysis of intrinsic (124) and mobile (186) resistance features. The resistome of CB and UB as well as resistances from genomes and plasmids differed significantly (Permutational Multivariate Analysis of Variance test: $n = 37$, pseudo-$F = 3.8$, $P = 0.004$ and pseudo-$F = 4.0$, $P = 0.002$; Fig. 5 and Supplementary Fig. 18). UB

showed more often mobile (10 vs 6%), transposable (36 vs 13%), replication (29 vs 10%) and slightly more virulence (6 vs 4%) factors or elements on their extracted plasmids than CB. Overall, interconnections of the resistome between genomes and extracted plasmids were very rare. Only a few genes encoding diverse efflux pumps (*pmrA* and *acrA*) could have been transferred between genomes and extracted plasmids of *Exiguobacterium sibiricum*, *Streptococcaceae* (both from UB), and *Stenotrophomonas maltophilia* (inside the cleanroom facility), respectively (Fig. 6), since they were detected in the same environment and/or recovered from similar genomes. However, the role they might have in resistance, particularly *acrA*, which forms the part of an intrinsic tripartite *Enterobacteriaceae* efflux pump, remains obscure. CB showed significantly higher abundance of elements involved in intrinsic resistance, including efflux pumps and stress-resistance determinants (e.g., as identified by LEfSe analysis, the multidrug efflux proteins *mexK* and *mexB*, and the catalase peroxidase-activating isoniazid *katG* in all CB environments). Besides built environment-specific profiles, species-specific patterns of the resistome were also observed; for instance, *smeA* in *S. maltophilia* (multidrug efflux) and *salA* in genomes of *Macrococcus caseolyticus* (possible resistances against lincosamides and streptogramins; Fig. 7a, b).

Further differences between CB and UB were also evident in terms of potentially conferred resistances against distinct drug classes. CBs were relatively enriched by resistances against fluoroquinolones ($W = 1705$, $P = 0.4$) and triclosan ($W = 1666$, $P = 0.02$) compared to UB. In turn, UBs were more representative of resistances against aminoglycoside ($W = 1842$, $P = 0.007$), diaminopyrimidine ($W = 1384.5$, $P = 0.7$), and macrolide-based antibiotics ($W = 1598.5$, $P = 1.0$; $P$ values from Wilcoxon signed-rank tests). Regarding their location, genes encoding beta-lactam,

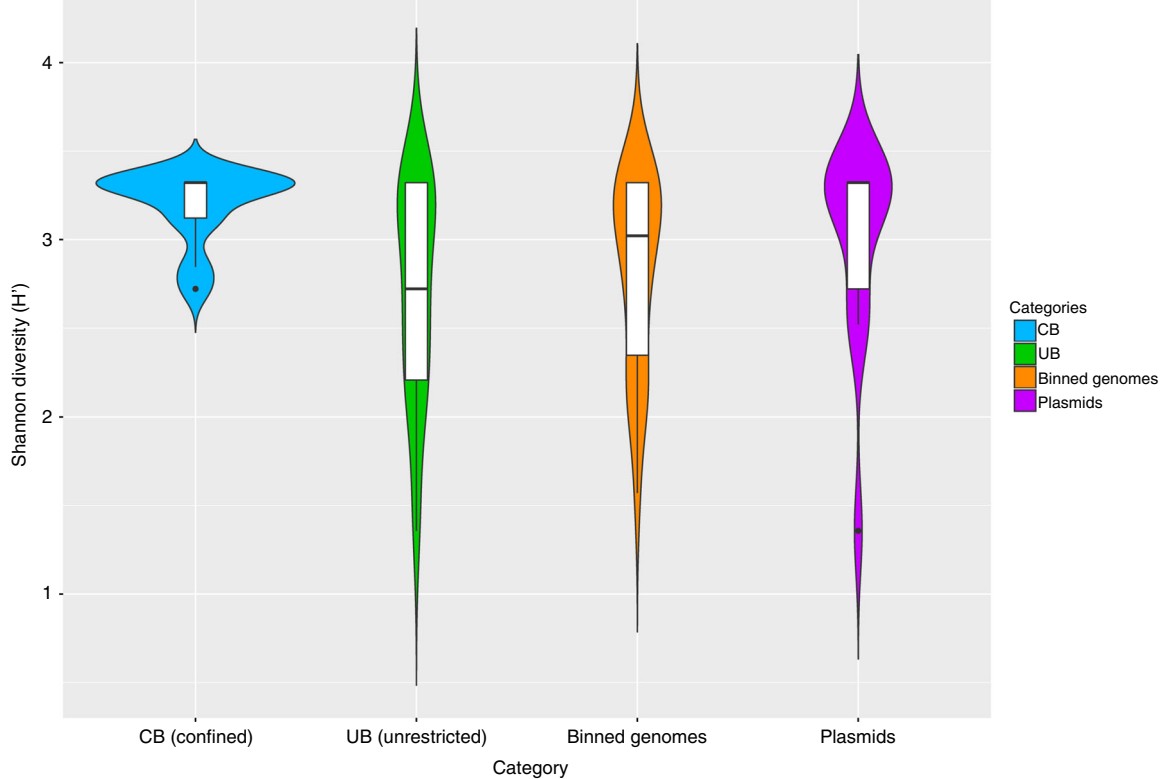

**Fig. 5** Diversity estimates of detected resistance features. Significant differences in Shannon diversity estimates of different resistance features (highest levels, level 3) of the CARD database inside CB (confined) and UB (unrestricted built environments) as well as on binned genomes and plasmids. Data were normalized (rarefied). CARD, Comprehensive Antibiotic Resistance Database

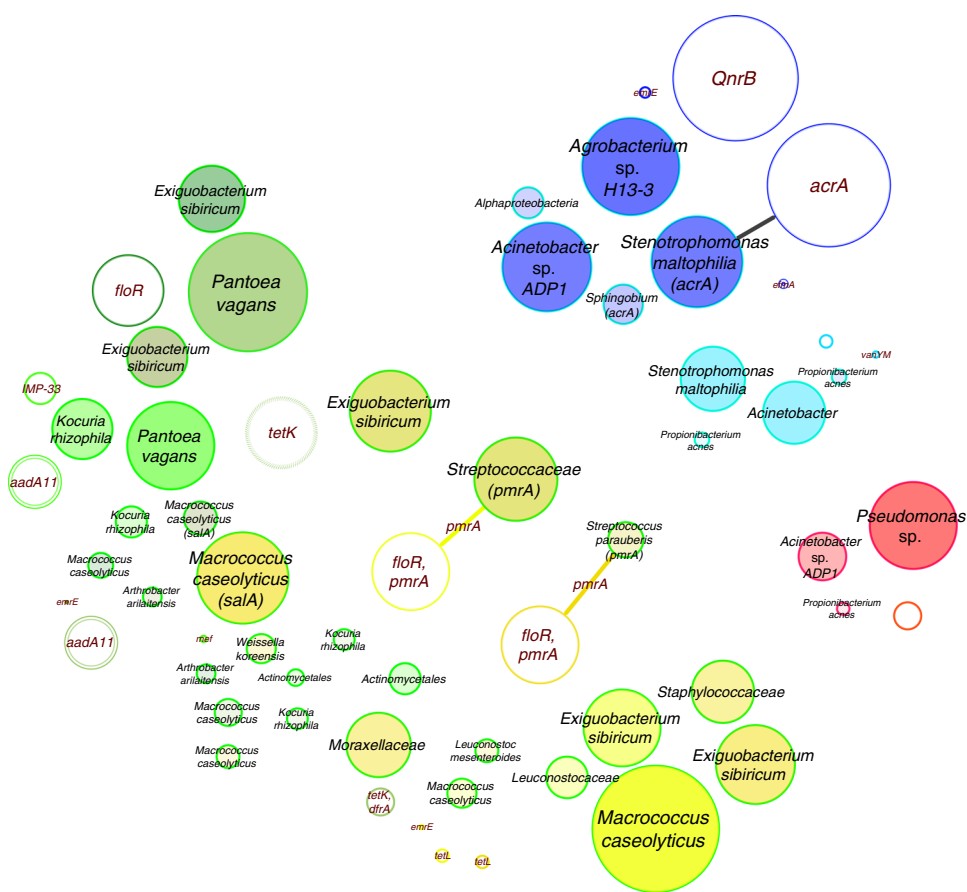

**Fig. 6** Resistance network of genomes and plasmids. Potentially transferred (edge-connected) resistance genes (CARD database) according to their presence/absence in binned genomes and plasmids inside the same built environment. Edge-weighted spring-embedded algorithms implemented in Cytoscape were used for visualizations. Filled circles represent genomes and empty circles, plasmids. Most abundant resistance genes were used for labeling and correlated to circle sizes. Colors are defined by respective built environments: blue (cleanroom facility); red (intensive care unit); dark green (public buildings); light green (public houses); yellow (private houses). CARD, Comprehensive Antibiotic Resistance Database

phenicol, and streptogramin resistance were more common in binned genomes, while extracted plasmids could mediate more resistances against fluoroquinolones, aminoglycosides, and diaminopyrimidines. Likewise, genomes of *Arthrobacter arilaitensis* showed many resistances against fluoroquinolones, while genomes assigned to *Acinetobacter* sp., *Pseudomonas* sp., and *Sphingobium* were rich in resistances against tetracyclines. *Stenotrophomonas maltophilia* harbored many resistances to both drug classes. On the contrary, more unspecific multidrug resistances were frequently common for *Staphylococcaceae*, *Macrococcus caseolyticus*, and *Exiguobacterium sibiricum*.

The core resistome of individually binned genomes was much more coherent (100% of core resistance genes in all genomes) than the core resistome of extracted plasmids or the different built environment categories (only 20–30% of core resistance genes in all plasmids). These data agree with the concept of intrinsic resistomes as a set of resistance genes present in all the (or most) members of a given species[12]. Hence, the core resistome of CB showed resistances against fluoroquinolones and aminocoumarins, while UB contained resistances to these antibiotics and additionally against tetracyclines and mupirocins.

As already shown for the composition of the microbiome, annotated resistance features were also used to build predictive models by supervised learning methods. Predictions were almost accurate if they were based on resistance genes (CB vs UB: overall accuracy = 91%) instead of microbial profiles (CB vs UB: overall accuracy = 92%). However, numerical environmental parameters like sea levels ($R = 0.64$, $P = 3.3 \times 10^{-3}$), temperature ($R = 0.46$,

$P = 0.09$), and microbial abundance ($R = 0.46$, $P = 0.06$) could not be predicted easily and showed only low model accuracies.

Resistance genes were further investigated in their genomic context (synteny). In most cases, antibiotic resistance genes were co-localized with other resistance genes especially on genomes retrieved from CB environments (mainly multidrug efflux transporter systems e.g., *acrA*, *acrB*, and *bepE*). In contrast, genomes from UB environments showed more often transcriptional regulators (e.g., *cymR* and *grpE*) and transposases (*tnpABC*) in close vicinity to annotated resistance genes. Despite the high frequency of transposase genes in the vicinity of resistance genes, no integron clusters could be detected. Resistance genes of genomes from CB environments were also significantly more often surrounded by a higher frequency of flanking repeats ($W = 12075$, $P = 0.02$). Potentially horizontally transferred genes (HGT) in regions of genome plasticity were identified by synteny breaks and the compositional bias between genomes of CB and UB and closely related genomes available in the MaGe database[13]. More potential HGT features (both mobility genes as well as tRNA hotspots) were detected in genomes from CB environments. However, higher proportions of HGT in CB were not significant.

In summary, a significant ($W = 110$, $P = 1.3 \times 10^{-7}$) reduction in microbial diversity on surfaces in CB by 50% was accompanied by a significant ($W = 202.5$, $P = 0.01$) increase of resistances by 20%, suggesting an enrichment of resistant microorganisms that displace the susceptible ones in these environments ($P$ values from Wilcoxon signed-rank tests).

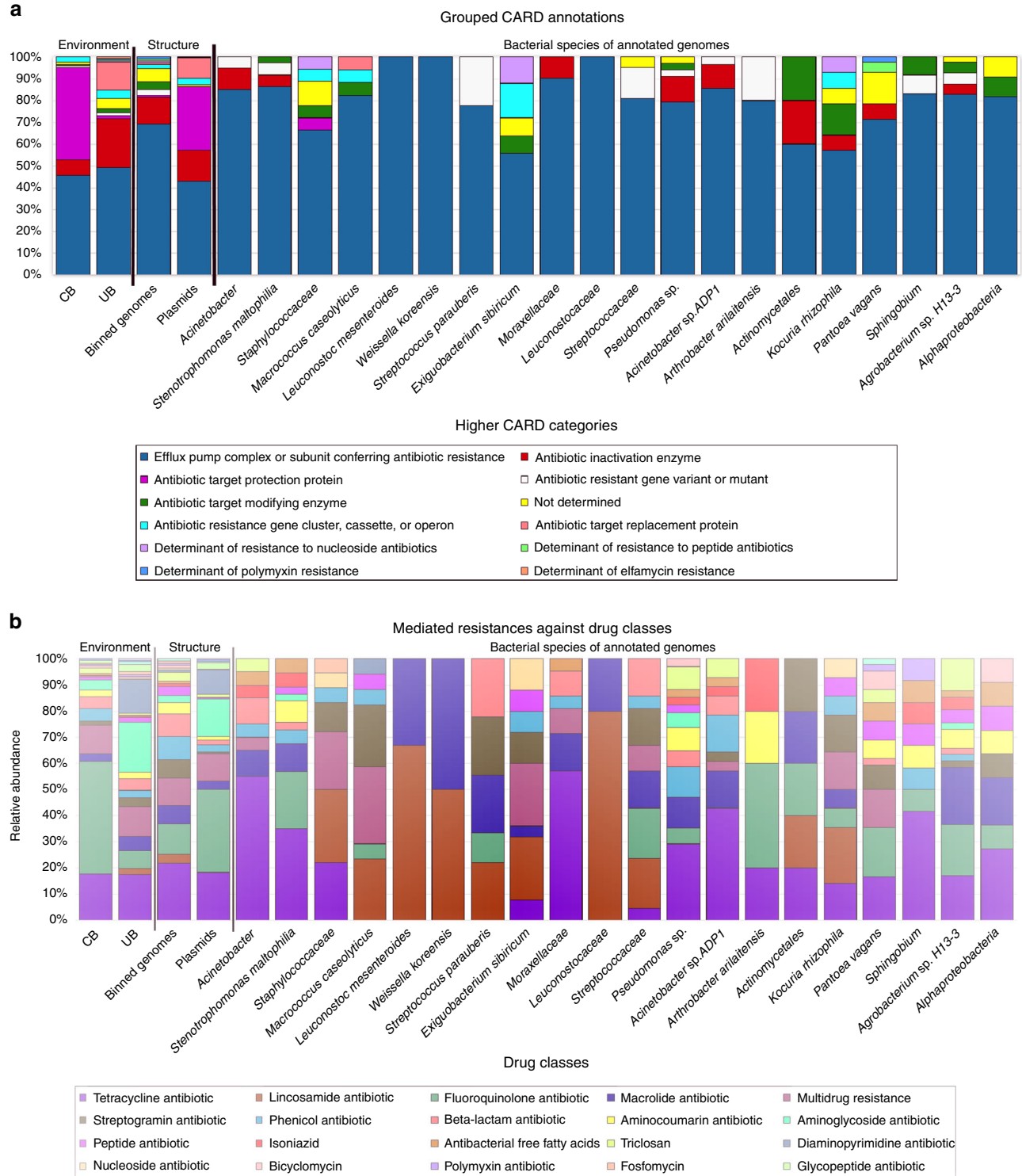

**Fig. 7** Proportion of CARD categories and drug classes. **a** Higher categories of the resistome according to CARD per environment (CB and UB), nucleotide structure (binned genomes and plasmids), and for individual binned genomes (referring to individual species). **b** Drug classes and their conferred resistance to them according to CARD per environment (CB and UB), nucleotide structure (binned genomes and plasmids), and for individual binned genomes (referring to individual species). CARD, Comprehensive Antibiotic Resistance Database

## Discussion

Our comparative analysis of deeply sequenced shotgun metagenomes and 16S rRNA gene amplicons revealed a clear microbial pattern on building surfaces characterized by different maintenance levels. While UBs were dominated by bacterial signatures commonly associated with the outdoor environment and processed food, CBs revealed a high abundance of sequences assigned to mainly human-associated bacteria, opportunistic pathogens, and only a low proportion of potentially beneficial bacteria (no potential pathogens; lower proportions of functions

associated with virulence, defense, and resistance). UBs were mainly colonized by robust Gram-positive bacteria with many functional capabilities to adapt to fluctuating conditions of the microclimate, UV radiation, and nutrient availability. Opposed to this, the constant moderate microclimate of CB and the strong anthropogenic influence selected for human-associated Gram-negative bacteria. Regular and strict cleaning procedures directed the microbiome to encode for functions associated with oxidative stress in combination with functions for membrane transport, secretion, and apoptosis to gather nutrients from a highly competitive nutrient-poor environment, a condition that was described as a wasteland for microbes[14]. Regular exposure to cleaning reagents and toxic compounds of these microbial populations were encountered by increased functional capabilities to degrade xenobiotics, geraniol, limonene, pinene, naphthalene, bisphenol, chlorocyclohexane, chlorobenzene, drug metabolism, and an overall higher level of virulence, disease, defense, and resistances. Investigated virulomes and resistomes underlined the strong impact of humans on microbiomes in the built environment. Virulence factors were more abundant for bacteria that need to survive in clean, nutrient-poor, and microbially controlled CB environments. Regarding antibiotic resistance, bacteria from CB tend to encode for a bigger diversity of genes involved in multidrug efflux, while bacteria from UB harbored more specific resistance features. The regular application of many different antibiotics and certain detergents might select such broad-spectrum resistance features in these microbial-controlled environments and also increase resistances against fluoroquinolones, triclosan, or elfamycins. Similar to this study, Lax and coworkers already reported a co-localization of different AMRs in close genomic context and the high proportion of multidrug efflux genes (e.g., *mexAC*) on hospital-associated surfaces[3].

Our resistome analysis covered not only the presence and abundance of detected resistance genes, but also their context in respective draft genomes as well as their potential to be horizontally transferable with known pathogens[2,5,15]. Besides this comprehensive analysis, the present study faces some limitations, such as the low sample size from CB environments, its focus on one sample type (floor samples), and the lack of metadata on specific administered antibiotics especially in the ICU at the time of sampling in contrast to other studies[3,4]. This low sample size was a consequence of the restricted access to the confined built environment setting of the ICU and the cleanroom facility as well as the low amount of biomass in these CB environments. Hence, the representativeness of the subsequent analysis is limited and also constrained our attempts to correlate and interpret microbial, virulence, or resistance compositions with environmental variables, as was shown in the study of Lax et al. in 2017[3]. Therefore, the general validity and impact of the presented results require additional confirmation by further studies.

Nevertheless, our study tried to substantiate the assessment of observed differences of the resistome between CB and UB by three aspects. First of all, the increased diversity of resistant features in CB was positively correlated with the number of potential pathogens. Secondly, we targeted potentially horizontally transferred genes in regions of genome plasticity, transposases, flanking repeats, and integron clusters as well as the resistome of plasmids to cover mobile genetic elements. And finally, we differentiated our analysis for intact microbial cells and determined the level of replication to emphasize the resistome inside metabolically active microbiota. Facing this differentiated analysis, the resistome of CB was more diverse, potentially mobile, and in increased contact to potential pathogens, but often less active and therefore harder to manipulate at the time of sampling. These aspects of CB in the presence of an overall decreased microbial diversity indicate an adverse anthropogenic influence. Many studies emphasize the role of microbial diversity to stabilize microbial communities and to act as a protection shield against the invasion of pathogens[16–18]. Hence, functional and compositional diversity can be considered as an unspecific, but universal, marker of ecosystem stability[19]. The current study, together with the previously published work[7], highlights that the loss of microbial diversity correlates with an increase of resistances, indicating that these populations might be burdened by antibiotic-resistant organisms. It is conceivable that the restoration of biodiversity may allow a decrease of antibiotic resistance.

However, while it is mandatory for cleanrooms to be almost free of microorganisms, other areas in hospitals or in private or public buildings do not need (or can) to be absent of microorganisms. Furthermore, cleaning for hygiene purposes does not imply the necessity to apply antimicrobial products that would propel adverse selection pressure on the resistome. Given that human interventions for reducing microbial load may cause a decline in microbial diversity, which is associated with the increase of antibiotic resistance in the microbiome, human exposure to almost sterile environments should be limited to operating rooms or particular industrial processes in cleanrooms. All other areas of life in the built environment could be enriched by a higher microbial diversity. One simple solution to increase microbial diversity is to increase the exchange of air with the outdoor environment by regular window ventilation. Or, as we proposed before, to introduce green plants, at least in close vicinity to confined areas[20–22]. Another step would be the active manipulation or 'biocontrol' of indoor and health-care environments[23,24]. Biocontrol is established already for other applications;[19] a first study indoors showed promising results through the application of *Bacillus* spores in a health-care setting[25].

Buildings are the main environment in which people spend their lives, share microbes, and where many diseases associated with anthropogenic activities may have their origin[26]. Moreover, microbial profiles are affected by microbial maintenance and building confinement[27]. However, an unselective removal and killing of many microbes in the built environment could have adverse health effects, since potent immune development may rely on microbial exposure[23,28–30]. In particular, such broad-spectrum selection mechanisms detected in CB environments are prone to damage the microbiome, which would lead to a loss of biodiversity and possibly to an accumulating effect over generations[17]. Hence, the confinement of a built environment should be limited to defined areas and special demands as indicated above. For all other built environments, building materials could be diverse to allow a higher microbial diversity[31]. Surface maintenance, such as cleaning, could be diversified, and the application of biocidal detergents can be limited to hot-spot locations and distinct timeframes. Also, in the end, the overall use of antimicrobials in buildings needs to be carefully considered.

The presence of highly diverse, stable, and beneficially designed microbiomes inside healthy buildings could result in lower exposures to resistances in the future. Since the overall heritability of microbiomes is much lower (up to 10-fold) than human beings acquire microbiomes by their behavior in, e.g. buildings[32], we are not condemned to lose millions of people due to antimicrobial resistances—instead it is time to reconsider our behavior in the built environment.

## Methods

**Environmental parameters and study design**. A variety of indoor environments different in their levels of microbial control, maintenance, and access were sampled during the same season of the year (spring). All these indoor environments featured different environmental parameters summarized in Table 1, which were suspected to contribute to the composition and function of their microbiome. More

**Table 1 Environmental parameters**

|  | Public buildings (L) | Public houses (P) | Private houses (F) | Intensive care unit (ICU) | Cleanroom gowning area (UR) | ISO8 cleanroom (CR) |
|---|---|---|---|---|---|---|
| Shotgun samples | 2 | 2 | 2 | 1 | 1 | 1 |
| Amplicon samples | 18 | 6 | 8 | 24 | 2 | 3 |
| Restriction level | Non |  | Partly | Moderate | High |  |
| Occupancy type | Uncontrolled |  | Restricted | Controlled |  |  |
| Surface material | Polymer, concrete, tiles, wood | Tiles, wood | Polymer, tiles, stone | Polymer, furnished wood, metals | Polymer (antistatic, dissipative epoxy resin) |  |
| Location | Floors |  |  | Floors, workplaces, medical devices | Floors |  |
| Building setting | Rural |  |  | Suburban | Urban |  |
| Maintenance | Conventional |  |  | Humidity, temperature | Particles, humidity, temperature, electromagnetics, electrostatics | Particles, humidity, temperature, electromagnetics, electrostatics |
| Cleaning | Mechanically (broom) | Natural soaps | All-purpose cleaners | All-purpose cleaners, surface disinfectants, sanitary cleaners | Isopropanol, Jaminal Plus, Klercide-CR | Isopropanol, Jaminal Plus, Klercide-CR, vapor phase $H_2O_2$ |
| Purpose | Education | Accommodation | Residence | Medical care | Changing garment | Spacecraft assembly |
| Characteristics | Reduced interaction of occupants with the outdoor environment and its influences | Sanitary area and kitchen included | Kitchen included, resident and dog (pet) | Medical care of patients, microbes, and viruses | Gowning area | HEPA air filtration, special garment, microbial control |
| Sampled surface ($m^2$) | 43 | 46 | 25 | <1 | 38 | 169 |
| Room size ($m^2$) | 43 | 46 | 25 | 75 | 38 | 169 |
| Room height (m) | 3 | 3 | 3 | 3 | 3 | 8 |
| Room volume ($m^3$) | 142 | 139 | 82 | 225 | 113 | 1349 |
| Humidity (%) | 62 | 65 | 60 | 32 | 55 | 55 |
| Temperature (°C) | 19 | 18 | 19 | 24 | 22 | 22 |
| Latitude | 53.950258 | 53.948030 | 53.953343 | 47.081353 | 45.079675 | 45.079675 |
| Longitude | 10.031095 | 10.026079 | 10.034461 | 15.465090 | 7.608334 | 7.608334 |
| Sea level (m) | 28 | 28 | 31 | 394 | 279 | 279 |

Selected metadata of sampled confined (CB) and unrestricted built environments (UB)

details about the study design and potential environmental influences and differences can be found in the Supplementary Methods (Supplementary Figs. 19–21 and Supplementary Tables 4, 5). Two types of floors of indoor environments with different cleanliness levels were investigated: UB, unrestricted buildings (public buildings, public and private houses) and CB, confined buildings (intensive care unit and cleanroom facility). The structure of the population and the whole metagenomic composition were investigated through 16S rRNA gene amplicon and shotgun metagenomic sequencing as described below.

**Sampling procedures**. Large-scale floor samples (defined by the size of each room) were collected to obtain high amounts of biomass (even from low-biomass environments like cleanrooms). In addition, floor samples were shown to have high diagnostic capacities of its occupants[4] as well as high proportions of antimicrobial resistances[3]. For this approach, sterile (autoclaved) and DNA-free (dry heat treatment) Alpha Wipes® (TX1009; VWR International GmbH, Vienna, Austria) were mounted in several layers separated by sterile, DNA-free foliage on a big swab (Swiffer® Sweeper® Floor Mop Starter Kit; Procter & Gamble Austria GmbH, Vienna, Austria) under a biohood. If necessary, wipes were remoistened by spraying polymerase chain reaction-grade water directly on the surface with a spray bottle. All instruments were chemically sterilized in several steps (all-purpose cleaner, Denkmit, dm-drogerie markt GmbH + Co. KG, Karlsruhe, Germany; 70% (w/v) ethanol, Carl Roth GmbH & Co. KG, Karlsruhe, Germany and Bacillol® plus, Bode Chemie GmbH, Hamburg, Germany). The remaining DNA was denatured with chlorine bleach (DNA away; Molecular Bio Products, Inc., San Diego, CA, USA) and UV light (254 and 366 nm; Kurt Migge GmbH, Heidelberg, Germany).

Samples were collected in a repetitive way, always starting from cleaner areas in each indoor environment (especially in cleanrooms according to their ISO classifications) to minimize the transfer of contaminants. Sampling was executed by the same person to guarantee a consistent sweeping pattern (horizontal, vertical, and diagonal sweeping motions) as well as a consistent uptake of particles and microbes. Samples were stored on blue ice and processed at the laboratory within 12 h after each sampling event. Samples from the ICU were already obtained and processed in a previous study[33], but are now included for comparative analysis.

**Sample processing, PMA treatment, and DNA extraction**. Samples were processed, concentrated, and treated with PMA prior to DNA extraction (more details are provided in Supplementary methods). PMA treatment of samples from high-biomass environments was performed as an additional control for potential DNA contaminants in used reagents and on sampling equipment. In addition, PMA treatment served as a proxy to evaluate the proportions of intact microbial cells and validate drawn conclusions on viable microbial cells in the dataset. The DNA-extraction method with the xanthogenate-SDS (XS) buffer was suitable for low-biomass environments; however for samples with higher biomass, an additional treatment with the Geneclean® Turbo Kit (MP Biomedicals, Heidelberg, Germany) according to manufacturer's instructions was necessary.

**Quantitative measures**. Bacterial abundance was investigated for most samples by quantitative polymerase chain reaction. Further details are specified in Supplementary methods.

**Shotgun metagenomics**. Total extracted DNA of all the samples was pooled into the categories of public buildings, public houses, private houses, ICU, and cleanroom and gowning area with a mean DNA amount of ∼ 10 μg and a mean DNA concentration of 149 ng/μl. After quality control, nine shotgun libraries were prepared by fragmentation and end repair of DNA with insert sizes of ∼ 300 bp. Sequencing was performed at Eurofins Genomics GmbH (Ebersberg, Germany) using an Illumina HiSeq 2500 instrument with 2 × 150 bp paired ends in the rapid run mode.

**16S rRNA gene amplicons and sequencing**. Amplicons targeting the 16S rRNA gene were generated with the barcoded primer pair 515f–806r[34,35] (primer sequences are listed in Supplementary Table 6). Further details can be found in Supplementary methods.

**Controls**. Negative controls were processed at each experimental step besides the actual samples. PMA treatment served as an additional quality control for free still-amplifiable background DNA in used reagents, equipment, and overall observations of low-biomass environments. Extraction controls and field blanks (samples of the background environment without any contact with the floor surface) were processed in parallel. Sequences of these control samples were subtracted from the normalized dataset during the bioinformatics analysis as described below.

**Bioinformatics**. Shotgun metagenomics: after the quality control of raw reads, sequencing adapters were removed from sequences and quality filtered according to phred score (> q35) as well as length filtered (min. 50 bp) by trimming from the 3' prime site (Supplementary Table 7). The whole analysis was conducted in a genome-centric approach focusing on assembly-based data (contigs, scaffolds, and bins). However, gene-centric analysis based on single reads served as a quality control for assembly-related artifacts throughout the analysis. These single reads were assigned using BLASTx search algorithm with default settings[36] against a custom-marker database (all COGs/NOGs in eggNOG 4.0[37], which can be found in 99% of all archaea, bacteria, and eukaryota) as well as against the NCBI non-redundant database (release 211.0 of December 2015). Annotations of all single reads were determined and analyzed with MEGAN (MEtaGenome ANalyzer)[38]. For the genome-centric approach, quality sequences were assembled with Ray Meta and a k-mer length of 31[39]. Assemblies were filtered according to the following parameters: minimum length 1500, minimum coverage 5, and read length 150. A summary of all filtered assemblies is provided in Supplementary Table 8. Afterwards, the filtered contigs were taxonomically classified with AMPHORA2[40] using the database of markers described above. For the visualization in Krona charts, the coverage ratios of respective contigs were considered to show relative abundances. Contigs were further binned through a genome-centric approach with CONCOCT[41] and MaxBin[42]. Binning quality of contigs was validated with CheckM[43] (Supplementary Table 2). Draft genomes in the range of 75–85% completeness and 2–25% contamination were considered to be suitable for downstream analysis. Contigs of each bin were re-annotated with AMPHORA2 and compared with publicly available genomes using RAST[9] and MaGe[13] to reveal ecologically relevant functional subsystems with special focus on pan-genomes (core genome and variable genome; MicroScope gene/protein families (MICFAMs) parameters: 80% amino acid identity, 80% alignment coverage), virulomes (running BLASTp on organism proteins against MicroScope, the virulence factor database VFDB[44] accessed in MaGe[13] with 60% identity and considering only best hits), and resistomes (CARD homologs and variants, v.1.1.2, RGI v.3.1.1[45]). Further details about the synteny analysis can be found in Supplementary methods. Predicted functional classifications of protein-coding genes were analyzed by annotation and comparative genomics in IMG[46] with GO terms (Gene Ontology Consortium, 2000), KEGG (Kyoto Encyclopedia of Genes and Genomes)[47], and SEED classifications[9]. Plasmids were extracted with Recycler[48] and annotated with KEGG, uniref90, and CARD (see Supplementary methods for more details). Binned genomes were compared with dRep[49] and replication rates were calculated with iRep[50]. Phenotype Investigation with Classification Algorithms (PICA) was used to predict the phenotypes of binned genomes (phendb.org)[51]. More details on the settings of used bioinformatic tools can be found in Supplementary Table 9. Bioinformatic analyses of the population structure based on 16S rRNA sequences are described in detail in Supplementary methods.

**Statistical information**. Statistical analyses were conducted in QIIME 1.9.1 and QIIME 2 versions 2017.10 and 2018.11[52] (calling respective R scripts) or directly in R[53] using the vegan package. Statistical tests included a comparison of categories, distances, distance matrices, core microbiomes and core functions, taxa summaries, co-occurrence patterns, correlations of metadata, a bioenv test (Supplementary Tables 10–12), and multivariate linear regression models. For nonparametric tests like multi-response permutation procedures, adonis, analysis of similarities, permutational multivariate analysis of variance, Kruskal–Wallis, Kolmogorow–Smirnow, Wilcoxon signed-rank test (Mann–Whitney $U$ test), Spearman rank correlations, distance-comparison box plots, Mantel correlograms, and Mantel tests, statistical significance was determined through 999 permutations. Distance-comparison box plots were calculated using a two-sided Student's two-sample $t$ test. All resulting $P$ values were Bonferroni corrected. PCoA plots were based on weighted unifrac metrics. NMDS was calculated from a Bray–Curtis distance matrix. Vectors of environmental variables shown in NMDS were calculated with the bioenv function based on Euclidean distances in R as were calculated ellipses per sample group. LEfSe[54] and microbiome: picking interesting taxonomic abundance analysis (microPITA) (http://huttenhower.sph.harvard.edu/micropita) were performed on Galaxy modules provided by the Medical University of Graz (https://galaxy.medunigraz.at/). Both tools were executed with default settings using an all-against-all strategy for the multi-class analysis for 16S rRNA gene amplicon datasets as well as CARD annotations and a one-against-all strategy for the Phylogenetic Investigation of Communities by Reconstruction of Unobserved States-predicted functions throughout the LEfSe analysis. Differential abundance of features was calculated with analysis of composition of microbiomes[55]. Sample metadata was predicted with random forest classification and regression models in QIIME 2[56].

**Reporting summary**. Further information on experimental design is available in the Nature Research Reporting Summary linked to this article.

## Data availability
All raw data from metagenomes, genomes, and 16S rRNA gene amplicons were deposited in the European Nucleotide Archive (https://www.ebi.ac.uk/ena) under project PRJEB27640. Processed shotgun reads are accessible from MG-RAST (http://metagenomics.anl.gov/) under project mgp10962. 16S rRNA gene amplicon reads were also deposited in Qiita[57] (https://qiita.ucsd.edu/) under study 10071. 16S rRNA gene amplicons from the ICU were published and deposited before[33].

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

## Acknowledgements

We like to thank Lisa Oberauner-Wappis for providing samples from the intensive care unit of the state hospital in Graz, Gerhard Kminek for facilitating sampling at the cleanroom facility operated by Thales Alenia Space in Turin, and Anna Auerbach and Simon Barczyk for sampling support. We also like to thank Isabelle Mahnert, Wolf-Gunthram Frhr. v. Schenck, and Theda Hatlapa for enabling sampling from public buildings and public and private houses inside the wildlife park in Eekholt, and Franziska Mahnert for preparing respective sampling locations. We also thank Tobija Glawogger for his help with sample processing and Javier Geijo for his support with phenotype predictions in PICA and helpful discussions about biostatistics with Slave Trajanoski from the Computational Bioanalytics Core Facility at the Medical University of Graz. This work was supported by a grant from the FWF (Austrian Science Fund) and the federal state government of Styria to G.B. (P29285-BBL). Sampling at the cleanroom facility was carried out under a contract with ESA and a subcontract with DLR given to C.M-E (ESTEC contract no. 4000103794/11/NL/EK) and for I.M. the work was supported by a grant from the European Research Council under the European Union's Horizon 2020 research and innovation program (grant agreement 640384).

## Author contributions

Study design: A.M., C.M-E. and G.B. Sampling and sample processing: A.M. Data analysis: A.M., M.Z., T.R., D.B., I.M. and J.L.M. Writing the manuscript: A.M., J.L.M. and G.B. All authors improved and approved the final manuscript.

## Additional information

**Competing interests:** The authors declare no competing interests.

