## [Peer Review File · Nature Communications]

Reviewers' comments:

Reviewer #1 (Remarks to the Author):

The manuscript by Mahnert et al. studies the microbial communities and resistome that were present in unrestricted and confined built environments. Shotgun metagenomic and 16S rRNA gene amplicon sequencing were applied. Comprehensive bioinformatics analyses were conducted including genome binning, genome reconstruction and plasmid reconstruction. This work is interesting especially the finding that highly maintained built environments led to a lower microbial diversity but higher resistance gene diversity. The results have strong implications on how to implement biocontrol in the built environments.

Major comments

1. The sample size of this study is relatively low, especially for metagenomic samples. Also, for some building types (e.g. hospital ICU), the samples were collected from one location only (i.e. one hospital). While accessing specialized facilities might be difficult, there needs to be a discussion about the limitations and representativeness of this study in the main text.
2. The quality of most of the figures in the Supplementary Materials needs to be improved.
3. Line 117. The use of PMA to evaluate intact microbial cells is useful when dealing with low biomass samples. It will be better to provide more information about the PMA methodology either in the main text or in the supplementary materials.
4. Line 164. Please indicate the criteria (e.g. % completeness, % contamination) used to evaluate whether a genome bin is suitable for downstream analysis.
5. Line 212 to 222. The results presented in this section need to have statistical support.
6. Line 231 to 250. Supplementary Figure S4 is not legible.
7. Line 251 to 261. The results presented in Figure 3A are not explained. Line 257 should only refer to Figure 3B.
8. Line 253 to 255. The number in the brackets needs to be explained. Is it statistical support? If yes, please indicate the method used.
9. Line 298 to 310. The completeness of the genomes can influence the interpretation of the core and strain-specific genomes. Information about completeness should be included.
10. Line 338 to 340. Please further explain this sentence and why the authors can draw the conclusion there is a shift in microbial composition.
11. Line 351 to 355. Are mobile genetic elements detected? This will make the transfer argument stronger.
12. Line 376. Please further explain the definition of coherent in the context of the results presented.
13. In the discussion, the authors should clarify whether there is a difference between cleaning with antimicrobial products versus cleaning for hygiene purposes. Line 441 to 442 could be misleading when the authors simply mentioned other built environments do not need to be microbially clean.

Reviewer #2 (Remarks to the Author):

Mahnert et al. used culture-independent approaches to investigate microbial diversity and antimicrobial resistance in a variety of built environments. The authors claim to have found trends in antimicrobial resistance profiles depending upon the environment sampled; however, I believe that their sampling strategy and analysis approach do not support their conclusions. Specifically, the authors took a non-quantitative approach (pooled samples) on a limited number of samples (9) and attempted to make quantitative conclusions. The conclusions are also overly broad for the number of samples represented. While the work described herein, especially the binning of microbial genomes, is potentially valuable to provide insights to antimicrobial resistance in built environments, the manuscript must be significantly reconstructed to make it acceptable for publication. Specifically, the manuscript should be re-written using a descriptive approach to better represent the research methods used. In addition, the manuscript would benefit by editing for clarity and English. Some other suggests/comments noted below for the authors' consideration.

L59,398 - How can surfaces be 'abiotic' if you're sampling them for microorganisms?

L141 – not clear how PMA treatment is itself a control

L338 – 'used'

L342 – using CARD, not 'according to' CARD

L402 – how were 'potentially beneficial bacteria' defined?

L416 – 'resistance'

L432 – not clear what 'harder to influence' means here?

L437-439 – need to specify 'resistance' to what; polluted is an awkward word choice; 'resistant organisms'

L442 – what is meant by 'microbially clean'? I sure hope hospitals and homes are 'microbially clean' (i.e. void of pathogens)

L460 – no evidence for this proposed accumulation effect

Reviewer #3 (Remarks to the Author):

This study presents a good analysis of some disparate built environments that presents further evidence to support the assertion that extremely clean environments are enriched for microbial genes associated with virulence, stress response and resistance. I have a few major and minor concerns.

Ln 55-57 and 62-63 – what about the Lax et al papers covering the Home and Hospital Microbiome studies (PMID:28539477; PMID:25170151), both of these cover shotgun metagenomic analysis of ARGs in home and clinical built environments? Surely these are relevant?

Ln 221-222 – did diversity estimates correlate at all with the proportion of intact cells?

Ln 232-233 "Hence, 232 proportions of bacteria versus eukaryota decreased significantly" I assume you mean metagenomic reads assigned to eukaryotes? What were these reads? There is considerable issues associated with assigning metagenomic reads to eukaryotic genomes due to the limited number of eukaryotic organisms in the database.

Ln 241-243 "However, this pattern changed in the private house and if sequences of intact cells were targeted by a PMA pre-treatment towards a predominance of Firmicutes (up to 55%)." – something is missing in this sentence, or it is written incorrectly. Are you saying that the ratio of Actino to Proteo was reserved in private houses (i.e. the 'pattern' mentioned), or that this was only the case if PMA treatment was not taken into consideration? This is confusing.

Ln 253-254 – “This bioenv analysis showed higher correlations of samples with latitude, 253 longitude and the sea level (0.9425)” because these variables defined the two groups CB Vs UB right? And obviously ICU is different from clean room – so these differences are likely nothing to do geographic location or elevation and everything to do with the fact that the samples are from different building types. This analysis is critically flawed, as you do not have examples of each building type in each location or elevation. Hence line 261-262 “ but also indicated that the difference between the ICU and all other sampled built 260 environments may be caused by biogeography or microclimate.” Is wrong. Is it not possible that temperature and relative humidity could also be affected by these different locations and elevations? Can you test for that – because otherwise, there is no way to use the metadata gathered to associate with differences in microbial community dynamics.

One thought, you could run ANCOM between the metadata variables – in R you can run ANCOM while controlling for co-variables, which might enable you to see if any of the variables are actually associated with significant differences when you control for geographic location and elevation, which are the key features which describe the different building types.

Just checking but did you run multi=hypothesis correction for all your statistical associations involving OTUs or genes. I didn't see it mentioned – if not, you need to do this and then correct all non-significant associations.

Ln 289-294 – “Genomes assigned to Exiguobacterium and Macrocooccus were commonly recovered from 289 diverse UB environments. Genomes of Arthrobacter and Janibacter were more specific for the 290 category of public buildings and public houses. Enhydrobacter, Kocuria and Pantoea were found 291 additionally in private houses together with Lactococcus and Staphylococcus. Leuconostoc marked 292 the transition from private houses to the ICU. And finally, genomes assigned to Propionibacterium, 293 Pseudomonas and Stenotrophomonas were characteristic to all CB environments (Figure 4). “ – why are these reported without significance? It would be easy to do this analysis based on read mapping density.

“Genomes of Acinetobacter from private houses, the ICU, the cleanroom and 301 its gowning area shared a core genome with 24 up to 39% of all CDS (coding DNA sequence). “ – do you mean that the pangenomes between these environments shared 24-39% of the CDSs between pairwise comparisons of the different buildings?

Ln 306-308 “Regarding all binned genomes, the ICU 306 environment showed the greatest density for its core genome (0.2% core CDS) compared to all 307 other sampled built environments” – what exactly are you trying to say here? Can you make this clearer please?

Ln 310 – “In CB the number of assigned functions to these categories almost doubled compared to UB.” Was this difference significant?

Ln 336-340 – again were these differences significant?

Ln 365 – “relatively enriched” does this mean significantly?

Ln 386-387 “However numerical environmental parameters like sea levels ($R = 0.64$, $P = 1.1 \times 10^{-3}$), temperature ($R = 0.46$, $P = 0.03$)” - does this suggest that the differences in these features between the ICU and cleanroom are not significant? Hence disrupting the relationship with elevation and location?

Ln 400. “outdoor environment and processed food,” – this is VERY hard to prove right?

Ln 438-439. “indicating that these populations might be polluted by resistance organism” – this language is inappropriate.

Ln 439-440. "It is conceivable that the restoration of biodiversity may allow a 439 decrease of antibiotic resistance." This concept needs a lot of caveats, for example, it is likely that increased microbial biomass associated with a broader array of bacterial diversity may reduce the ability to detect taxa that are resistant to these environmental constraints, but what might the implications of this be?

Ln 441 "mandatory for cleanrooms to be almost void" is there a metric of 'voidness' – how is it mandatory?

One thing that feels missing is an appropriate analysis of the genomic context – synteny – of the resistance genes, i.e. are ARGs co-associated with other resistance-genes, e.g. those to moisture deprivation, or nutrient acquisition. In Lax et al 2017 Sci Trans Med they suggested that the genomes of microbes that were enriched in patient rooms were enriched in co-associated genes encoding ARGs and host-invasion genes. So what's going on in these organisms.

Response to Referees / Reviewers

Dear Reviewers,

thank you for all your valuable comments! We invested a lot of effort and utilized the entire granted timespan of three months to substantially revise our manuscript. We covered each comment raised and performed extensive additional analysis of our dataset. In particular, we addressed: a) the relatively low sample size and sampling strategy by discussing limitations and representativeness of our study in the main text; b) revised and added appropriate statistical analysis as well as tried to support observed associations with further tests; c) expanded the analysis of detected resistance genes in their genomic context (synteny).

We hope that you will appreciate these substantial revisions!

Sincerely yours

Alexander & coauthors

Reviewers' comments and Author replies (A:)

Reviewer #1 (Remarks to the Author):

The manuscript by Mahnert et al. studies the microbial communities and resistome that were present in unrestricted and confined built environments. Shotgun metagenomic and 16S rRNA gene amplicon sequencing were applied. Comprehensive bioinformatics analyses were conducted including genome binning, genome reconstruction and plasmid reconstruction. This work is interesting especially the finding that highly maintained built environments led to a lower microbial diversity but higher resistance gene diversity. The results have strong implications on how to implement biocontrol in the built environments.

Major comments

1. The sample size of this study is relatively low, especially for metagenomic samples. Also, for some building types (e.g. hospital ICU), the samples were collected from one location only (i.e. one hospital). While accessing specialized facilities might be difficult, there needs to be a discussion about the limitations and representativeness of this study in the main text.

A: We are aware of the low sample size in our study and included a discussion about the limitations and representativeness of this study in the discussion. Nevertheless, significant associations were discovered and in comparison to the literature we investigated taxonomically representative samples.

“Beside this comprehensive analysis the present study faces some limitations, such as the low sample size from CB environments, its focus on one sample type (floor samples), and the lack of metadata on specific administered antibiotics especially in the ICU at the time of sampling in contrast to other studies ^{3,4}. This low sample size was a consequence of the restricted access to the confined built environment setting of the ICU and the cleanroom facility as well as the low amount of biomass in this CB environments. Hence, the representativeness of the subsequent analysis is limited and also constrained our attempts to correlate and interpret microbial, virulence or resistance compositions with environmental variables as it was shown in the study of Lax et al., in 2017 ³. Therefore the general validity and impact of the presented results require additional confirmation by further studies.”

2. The quality of most of the figures in the Supplementary Materials needs to be improved.

A: We improved the quality (resolution and size) of most figures in the Supplementary Material (maybe quality is still reduced during file conversions – if so, they could be provided as svg files at a later stage).

3. Line 117. The use of PMA to evaluate intact microbial cells is useful when dealing with low biomass samples. It will be better to provide more information about the PMA methodology either in the main text or in the supplementary materials.

A: As suggested, the PMA treatment is now described with more details in the supplementary material.

“PMA treatment of samples:

An aliquot of selected biological samples and spot tests of used reagents were treated with propidium monoazide (PMA) according to manufacturer instructions (GenIUL, S.L., Terrassa, Spain). Samples were treated with a final concentration of 50 μ M PMA for 30 min. in the dark. Afterwards treated and non-treated samples were exposed in parallel to the PhAST blue-Photo activation system for tubes (GenIUL, S.L., Terrassa, Spain) for 15 min.”

4. Line 164. Please indicate the criteria (e.g. % completeness, % contamination) used to evaluate whether a genome bin is suitable for downstream analysis.

A: Now we provide our quality criteria for binned genomes.

“Draft genomes in the range of 75 – 85% completeness and 2 – 25% contamination were considered to be suitable for downstream analysis.”

5. Line 212 to 222. The results presented in this section need to have statistical support.

A: Beside Students *t*-tests we performed Spearman's rank correlations for increased confinement and reduced microbial diversity as well as correlations between diversity estimates and the proportion of intact cells. (Spearman's rank correlation ρ , correlation coefficient: -0.8783, *P*-value = 0.02131 and correlation coefficient: 0.2, *P*-value = 0.4)

6. Line 231 to 250. Supplementary Figure S4 is not legible.

A: Panels of Figure S4 are now displayed as individual Supplementary figures (S6, S7, S8 and S9) to make them more legible.

7. Line 251 to 261. The results presented in Figure 3A are not explained. Line 257 should only refer to Figure 3B.

A: This is correct. It is now explained in the main text, but was moved to the supplementary (Figure 3A is now Supplementary Figure S10 and Figure 3B is now Figure 3).

“The core 16S rRNA gene microbial profile was visualized in a core OTU network (Supplementary Figure S7). This analysis indicated a high proportion of shared OTUs assigned to Acinetobacter and Staphylococcus as well as a bigger overlap of samples from the clean room facility and unrestricted buildings compared to the core of samples from the ICU environment.”

8. Line 253 to 255. The number in the brackets needs to be explained. Is it statistical support? If yes, please indicate the method used.

A: Yes the number in brackets has statistical support, it is a correlation coefficient (0 – no correlation, 1 highest correlation). It is the maximally rank-correlated variable (or the best variable combination, ρ_w).

See:

KR Clarke and M Ainsworth. A method of linking multivariate community structure to environmental variables. Marine ecology progress series, pages 205–219, 1993.

for more details.

9. Line 298 to 310. The completeness of the genomes can influence the interpretation of the core and strain-specific genomes. Information about completeness should be included.

A: We now include information about the median completeness (94%) and contamination (20%) of these genomes in the manuscript.

“Genomes assigned to the genus of Acinetobacter (median completeness 94%, median contamination 20%) were highly prevalent and ubiquitous in all sampled built environments.”

10. Line 338 to 340. Please further explain this sentence and why the authors can draw the conclusion there is a shift in microbial composition.

A: Observations on the virulome are based on our binned draft genomes. In this paragraph we characterized CB and UB environments according to this subset of obtained genomes. As indicated before we detected a distinct microbial profile for CB and UB respectively. This information refers to the stated “shift in microbial composition”. Further on as the detected virulence factors used to be chromosomally encoded we see a succession of different compositions also for the different sets of virulence factors. We tried to simplify this sentence so that it does not combine two distinct observations (microbial composition and composition of virulence factors) in one phrase.

“Hence, chromosomally encoded bacterial virulence in CB and UB was likely associated to its distinct microbial profiles. “

11. Line 351 to 355. Are mobile genetic elements detected? This will make the transfer argument stronger.

A: Above we describe a higher proportion of mobile and transposable elements on plasmids. In addition we included a separate paragraph about the synteny of chromosomally detected resistance genes. This analysis showed that all resistance genes were associated to flanking repeats and significantly ($W = 12075$, $P = 0.02$) higher repeat frequency was observed for resistance genes from CB environments. Regions of genomic plasticity were screened for potentially horizontally transferred genes (HGT, both mobile genes and tRNA hotspots). However, despite the frequency of transposase genes no integron clusters could be detected in our draft genomes. More information can be found below:

“Resistance genes were further investigated in their genomic context (synteny). In most cases antibiotic resistance genes were co-localized with other resistance genes especially on genomes retrieved from CB environments (mainly multidrug efflux transporter systems e.g. *acrA*, *acrB* and *bepE*). In contrast, genomes from UB environments showed more often transcriptional regulators (e.g. *cymR* and *grpE*) and transposases (*tnpABC*) in close vicinity to annotated resistance genes. Despite the high frequency of transposase genes in vicinity to resistance genes, no integron clusters could be detected. Resistance genes of genomes from CB environments were also significantly more often surrounded by a higher frequency of flanking repeats ($W = 12075$, $P = 0.02$). Potentially horizontally transferred genes (HGT) in regions of genome plasticity (RGP) were identified by synteny breaks and compositional bias between genomes of CB and UB and closely related genomes available in the MaGe database 20. More potential HGT features (both mobility genes as well as tRNA hotspots) were detected in genomes from CB environments. However, higher proportions of HGT in CB were not significant.”

12. Line 376. Please further explain the definition of coherent in the context of the results presented.

A: By the term coherent we mean the proportion of resistance features in a certain fraction of (genomes or plasmids). More coherence just means a bigger fraction of core resistance features was observed in distinct set of genomes or plasmids (“a more conserved core”).

We quoted this definition in the manuscript: “(100% of core resistance genes in all genomes)” and “(only 20-30% of core resistance genes in all plasmids)”.

13. In the discussion, the authors should clarify whether there is a difference between cleaning with antimicrobial products versus cleaning for hygiene purposes. Line 441 to 442 could be misleading when the authors simply mentioned other built environments do not need to be microbially clean.

A: Thank you for this valuable comment! We added the following sentence to make this clear: “Furthermore, cleaning for hygiene purposes do not imply the necessity to apply antimicrobial products which would propel adverse selection pressure on the resistome.”

Reviewer #2 (Remarks to the Author):

Mahnert et al. used culture-independent approaches to investigate microbial diversity and antimicrobial resistance in a variety of built environments. The authors claim to have found trends in antimicrobial resistance profiles depending upon the environment sampled; however, I believe that their sampling strategy and analysis approach do not support their conclusions. Specifically, the authors took a non-quantitative approach (pooled samples) on a limited number of samples (9) and attempted to make quantitative conclusions. The conclusions are also overly broad for the number of samples represented. While the work described herein, especially the binning of microbial genomes, is potentially valuable to provide insights to antimicrobial resistance in built environments, the manuscript must be significantly reconstructed to make it acceptable for publication. Specifically, the manuscript should be re-written using a descriptive approach to better represent the research methods used.

In addition, the manuscript would benefit by editing for clarity and English. Some other suggests/comments noted below for the authors' consideration.

A: We substantially revised the manuscript and added new data and statistics. We agree that our study would benefit from larger sample sizes. However, selected confined built environments were not only difficult to access, but also challenging due to the low biomass; instead of analyzing more samples, we decided to perform deeply sequenced shotgun metagenomics without DNA amplification step that could introduce bias and support the shotgun data (read- and genome centric) with 16S rRNA gene amplicons. This strategy resulted in a higher effort to get samples and to process them. Nevertheless, we would have been delighted to support our conclusions with a larger sample size, but the scope of our study did not allow further analysis for now. We included a paragraph about these obvious limitations in the discussion section:

“Beside this comprehensive analysis the present study faces some limitations, such as the low sample size from CB environments, its focus on one sample type (floor samples), and the lack of metadata on specific administered antibiotics especially in the ICU at the time of sampling in contrast to other studies ^{3,4}. This low sample size was a consequence of the restricted access to the confined built environment setting of the ICU and the cleanroom facility as well as the low amount of biomass in this CB environments. Hence, the representativeness of the subsequent analysis is limited and also constrained our attempts to correlate and interpret microbial, virulence or resistance compositions with environmental variables as it was shown in the study of Lax et al., in 2017 ³. Therefore the general validity and impact of the presented results require additional confirmation by further studies.”

Instead of completely re-writing the manuscript, we extensively modulated several parts of the main text as this was much more in accordance with the comments of the other reviewers. However, we tried to add clarity to the whole manuscript.

L59,398 - How can surfaces be 'abiotic' if you're sampling them for microorganisms?

A: The surface material microbes can attach to is certainly abiotic, as the use of biotic surfaces in cleanrooms etc. is not allowed according to ISO classifications. Nevertheless, as you indicated this term could be misleading. Therefore, we removed it as suggested.

L141 – not clear how PMA treatment is itself a control

A: Lab reagents are sterile, but often still contaminated by DNA fragments. For molecular assays targeting low-biomass environments like cleanrooms it can be of advantage to mask this DNA with a PMA treatment for further downstream processes. Therefore, a PMA treatment can control for free still amplifiable "background" DNA in your samples. We tried to make this clear in the M&M section: "PMA treatment served as an additional quality control for free still amplifiable background DNA in used reagents, equipment and overall observations of low-biomass environments".

L338 – 'used'

A: Sentence was revised.

L342 – using CARD, not 'according to' CARD

A: Revised as suggested.

L402 – how were 'potentially beneficial bacteria' defined?

A: According to their taxonomy (e.g. BugBase predictions of potential pathogens) and lower proportions of functions associated to virulence, defense and resistance.

"...and only a low proportion of potentially beneficial bacteria (no potential pathogens, lower proportions of functions associated to virulence, defense and resistance)."

L416 – 'resistance'

A: Revised as suggested.

L432 – not clear what ‘harder to influence’ means here?

A: We assume that it is easier to treat or manipulate microorganisms when they are viable and show an active metabolism in contrast to microbes that outlast as e.g. spores in an environment.

We revised this part to make it clear and simple: “harder to manipulate”

L437-439 – need to specify ‘resistance’ to what; polluted is an awkward word choice; ‘resistant organisms’

A: We revised this part as suggested: “...might be burdened by antibiotic resistant organisms”

L442 – what is meant by ‘microbially clean’? I sure hope hospitals and homes are ‘microbially clean’ (i.e. void of pathogens)

A: We revised this part and tried to make it clear by specifying it: “do not need (or can) be absent of microorganisms.”

L460 – no evidence for this proposed accumulation effect

A: Correct! We did not investigate accumulation effects in our study, but assume that they could take place as mentioned by Blaser et al., 2016. We changed this phrase to “and possibly to an accumulating effect over generations”.

Reviewer #3 (Remarks to the Author):

This study presents a good analysis of some disparate built environments that presents further evidence to support the assertion that extremely clean environments are enriched for microbial genes associated with virulence, stress response and resistance. I have a few major and minor concerns.

Ln 55-57 and 62-63 – what about the Lax et al papers covering the Home and Hospital Microbiome studies (PMID:28539477; PMID:25170151), both of these cover shotgun metagenomic analysis of ARGs in home and clinical built environments? Surely these are relevant?

A: You are right. We cite these papers in the introduction, materials & methods and also compared them in the discussion section.

“An exception were the studies by Lax and coworkers, who investigated AMR not only in a hospital setting ³, but also in private homes ⁴.”

“In addition, floor samples were shown to have high diagnostic capacities of its occupants ⁴ as well as high proportions of antimicrobial resistances ³.”

“Similar to this study, Lax and coworkers already reported a co-localization of different AMRs in close genomic context and the high proportion of multidrug efflux genes (e.g. mexAC) on hospital-associated surfaces ³.”

“Beside this comprehensive analysis the present study also suffers from obvious limitations such as the low sample size from CB environments, its focus on one sample type (floor samples), and the lack of metadata on specific administered antibiotics especially in the ICU at the time of sampling in contrast to other studies ^{3,4}. This low sample size was a consequence of the restricted access to the confined built environment setting of the ICU and the cleanroom facility as well as the low amount of biomass in this CB environments. Hence, the representativeness of the following study is limited and also constrained our attempts to correlate and interpret microbial, virulence or resistance compositions with environmental variables as it was shown in the study of Lax et al., in 2017 ³. Therefore the general validity and impact of the presented results need to be verified by further studies in the future.”

Ln 221-222 – did diversity estimates correlate at all with the proportion of intact cells?

A: We calculated correlations of the bacterial abundances with the proportion of intact cells and level of confinement. For both cases we could not determine any correlation (correlation coefficient 0.2, -0.2 ;P = 0.4, 0.2 respectively).

Ln 232-233 “Hence, 232 proportions of bacteria versus eukaryota decreased significantly”
I assume you mean metagenomic reads assigned to eukaryotes? What were these reads?
There is considerable issues associated with assigning metagenomic reads to eukaryotic genomes due to the limited number of eukaryotic organisms in the database.

A: Here we talk about metagenomics reads assigned to eukaryotes according to NCBI nr. Mainly these reads were assigned to Metazoa (31% e.g. Homo sapiens up to 19% in CB), Fungi (1% e.g. Malassezia globosa up to 0.3%) and plants (Streptophyta 2%). This is a common profile we regularly see also in other confined built environments and underlines the impact of humans in such indoor spaces. We tried to make this clear by revising the sentence as follows:

“Different categories of sampled built environments could be characterized by distinct compositions of the metagenomics reads even on superkingdom level (Supplementary Fig. S6). Hence, proportions of bacteria versus eukaryota (mainly sequences assigned to humans) decreased significantly...”

Ln 241-243 “However, this pattern changed in the private house and if sequences of intact cells were targeted by a PMA pre-treatment towards a predominance of Firmicutes (up to 55%).” – something is missing in this sentence, or it is written incorrectly. Are you saying that the ratio of Actino to Proteo was reserved in private houses (i.e. the ‘pattern’ mentioned), or that this was only the case if PMA treatment was not taken into consideration? This is confusing.

A: Sorry for the confusion. By pattern we mean the dominance of sequences assigned to Actinobacteria and Proteobacteria. This pattern changed in the environment of the private house. Here we detected a dominance of sequences assignable to Firmicutes. And finally the proportion of Firmicutes in our samples also raised if they were treated with PMA compared to untreated samples. We revised the sentence and hope that it is now easier for the reader to follow this result.

“On phylum level, public buildings and public houses were dominated by sequences of Actinobacteria (up to 50%) and Proteobacteria (~ 21%). In the private house the proportion of Firmicutes raised up to 55%. Likewise the proportion of Firmicutes was also higher after masking the DNA of compromised cells with PMA.”

Ln 253-254 – “This bioenv analysis showed higher correlations of samples with latitude, 253 longitude and the sea level (0.9425)” because these variables defined the two groups CB Vs UB right? And obviously ICU is different from clean room – so these differences are likely nothing to do geographic location or elevation and everything to do with the fact

that the samples are from different building types. This analysis is critically flawed, as you do not have examples of each building type in each location or elevation. Hence line 261-262 “ but also indicated that the difference between the ICU and all other sampled built 260 environments may be caused by biogeography or microclimate.” Is wrong. Is it not possible that temperature and relative humidity could also be affected by these different locations and elevations? Can you test for that – because otherwise, there is no way to use the metadata gathered to associate with differences in microbial community dynamics.

One thought, you could run ANCOM between the metadata variables – in R you can run ANCOM while controlling for co-variables, which might enable you to see if any of the variables are actually associated with significant differences when you control for geographic location and elevation, which are the key features which describe the different building types

A: You are right. We tried to test and differentiate between associations of microclimate and location for our microbiome profile. We used MASLin, balances in gneiss and its regression summary and linear mixed effect models. However, after consulting a bio-statistician we are convinced that we cannot really resolve these connected variables with the sample size we have. We removed the statement (as it was not a central statement of our manuscript) and further revised this sentence as:

“However, associations of the microbiome with environmental variables like biogeography or microclimate could not be further supported or differentiated due to confounding variables (see Supplementary information).”

We provide more information about our attempts to differentiate the associations of these variables in the supplementary.

“Verification of bioenv results:

Associations of the microbiome with microclimate or location specific variables could not be further distinguished. MaAsLin was able to define specific taxa (distinct sets, only 6 of 82 were overlapping) for microclimate and location specific variables (e.g. microclimate: *Bauldia*, *Gaiella* and *Intrasporangium*; location: *Commensalibacter*, *Chlorocromatium*; both: *Iamia*, *Rubrobacter*). However, regression models using balances in gneiss showed that microclimate and location dependent variables contributed to similar proportions (~2%) to the total explained community variation (~70%). Moreover over-fitting of the model could not be ruled out (in 4 out of 6 cross-validations the prediction accuracy was higher than the within model error). Finally, linear mixed effect models were used to test if microbial composition changed over microclimate or location in response to confinement and architecture (room size). This analysis showed that microbial composition was not significantly impacted by these selected variables. Hence, we concluded that environmental variables of the microclimate and the location were confounded in

our sample design and were not appropriate to tell if the microclimate or the location has a bigger impact on the microbial composition.”

We also used this problem to indicate the limitations of our study in the discussion section.

“Hence, the representativeness of the subsequent analysis is limited and also constrained our attempts to correlate and interpret microbial, virulence or resistance compositions with environmental variables as it was shown in the study of Lax et al., in 2017 ³. Therefore the general validity and impact of the presented results require additional confirmation by further studies.”

Just checking but did you run multi-hypothesis correction for all your statistical associations involving OTUs or genes. I didn't see it mentioned – if not, you need to do this and then correct all non-significant associations.

A: We did multi-test corrections (Bonferroni correction). See M&M section on statistical information. However, after double checking we realized that some P-values were missed. Now all should be corrected.

“All resulting P-values were Bonferroni-corrected”.

Ln 289-294 – “Genomes assigned to Exiguobacterium and Macrocooccus were commonly recovered from 289 diverse UB environments. Genomes of Arthrobacter and Janibacter were more specific for the 290 category of public buildings and public houses. Enhydrobacter, Kocuria and Pantoea were found 291 additionally in private houses together with Lactococcus and Staphylococcus. Leuconostoc marked 292 the transition from private houses to the ICU. And finally, genomes assigned to Propionibacterium, 293 Pseudomonas and Stenotrophomonas were characteristic to all CB environments (Figure 4). “ – why are these reported without significance? It would be easy to do this analysis based on read mapping density.

A: We did this analysis for each taxon and now show all Bonferroni corrected P-values.

Genomes assigned to Exiguobacterium ($V = 0$, $P = 2.2 \times 10^{-11}$) and Macrocooccus ($V = 0$, $P = 1.0$) were commonly recovered from diverse UB environments. Genomes of Arthrobacter ($V = 465.5$, $P = 2.9 \times 10^{-15}$) and Janibacter ($V = 0$, $P = 0.3$) were more specific for the category of public buildings and public houses. Enhydrobacter ($V = 0$, $P = 1.0$), Kocuria ($V = 0$, $P = 8.3 \times 10^{-4}$) and Pantoea ($V = 225$, $P = 1.2 \times 10^{-9}$) were found additionally in private houses together with Lactococcus ($V = 9$, $P = 1.0$) and Staphylococcus ($V = 3445$, $P = 0.01$). Leuconostoc ($V = 169$, $P = 0.9$) marked the transition from private houses to the ICU. And finally, genomes assigned to Propionibacterium ($V = 2697$, $P = 0.01$), Pseudomonas ($V = 133530$, $P = 2.9 \times 10^{-15}$)

and *Stenotrophomonas* ($V = 97.5$, $P = 0.07$) were characteristic to all CB environments (P -values from Wilcoxon signed rank tests).

“Genomes of *Acinetobacter* from private houses, the ICU, the cleanroom and 301 its gowning area shared a core genome with 24 up to 39% of all CDS (coding DNA sequence). “ – do you mean that the pangenomes between these environments shared 24-39% of the CDSs between pairwise comparisons of the different buildings?

A: Not exactly. It is the proportion of core CDS (CDS of the pangenome) to all CDS in a specific genome (pairwise comparisons). We tried to make this clear with the following details in the text.

“...(proportion of core coding DNA sequences of all coding DNA sequences in a genome)...”

Ln 306-308 “Regarding all binned genomes, the ICU 306 environment showed the greatest density for its core genome (0.2% core CDS) compared to all 307 other sampled built environments” – what exactly are you trying to say here? Can you make this clearer please?

A: Compared to all other built environment categories, binned genomes from the ICU showed the highest grade of similarity to each other. Again we give more details in the text.

“...(highest grade of similarity)...”

Ln 310 – “In CB the number of assigned functions to these categories almost doubled compared to UB.” Was this difference significant?

A: No this difference was not significant ($V = 3$, $P = 0.4$).

Ln 336-340 – again were these differences significant?

A: These differences were not significant. ICU ($W = 11$, $P = 1$), private house ($W = 65.5$, $P = 0.09$), public house ($W = 30$, $P = 1$) and public buildings ($W = 7$, $P = 1$). We now include a statement in the manuscript.

“However, differences in proportions were not significant.”

Ln 365 – “relatively enriched” does this mean significantly?

A: The mentioned drug classes were only partly significantly associated to CB or UB environments. Fluoroquinolones ($W = 1705$, $P = 0.4$), triclosan ($W = 1666$, $P = 0.02$), aminoglycosides ($W = 1842$, $P = 0.007$), diaminopyrimidines ($W = 1384$, $P = 0.7$) and macrolide-based antibiotics ($W = 1598.5$, $P = 1.0$). All P -values were Bonferroni corrected.

We now include these P -values also in the text: “CB were relatively enriched by resistances against fluoroquinolones ($W = 1705$, $P = 0.4$) and triclosan ($W = 1666$, $P = 0.02$) compared to UB. In turn UB were more representative for resistances against aminoglycoside ($W = 1842$, $P = 0.007$), diaminopyrimidine ($W = 1384$, $P = 0.7$) and macrolide-based antibiotics ($W = 1598.5$, $P = 1.0$; P -values from Wilcoxon signed rank tests). “

Ln 386-387 “However numerical environmental parameters like sea levels ($R = 0.64$, $P = 1.1 \cdot 10^{-3}$), temperature ($R = 0.46$, $P = 0.03$)” - does this suggest that the differences in these features between the ICU and cleanroom are not significant? Hence disrupting the relationship with elevation and location?

A: We do not think so. This observation only tells us that resistance features are less suitable to predict these environmental parameters compared to different taxa. There might be many reasons, but we simply assume that the number of features could play a role here (lower amount of resistance features compared to microbial species to train the predictive model).

Ln 400. “outdoor environment and processed food,” – this is VERY hard to prove right?

A: We have no prove for that. We simply base our assumptions on the literature and obvious connections between the ecology of a species and the environment that we sampled (e.g. associate species to processed food because their sequences where obtained from a kitchen). We think this hypothetical origin of the detected microbes is sufficiently indicated by the phrase: “bacterial signatures commonly associated to”

Ln 438-439. “indicating that these populations might be polluted by resistance organism” – this language is inappropriate.

A: You are right. We revised this part as follows:

“...indicating that these populations might be burdened by antibiotic resistant organisms.”

Ln 439-440. “It is conceivable that the restoration of biodiversity may allow a 439 decrease of antibiotic resistance.” This concept needs a lot of caveats, for example, it is likely that increased microbial biomass associated with a broader array of bacterial diversity may reduce the ability to detect taxa that are resistant to these environmental constraints, but what might the implications of this be?

A: The implication would be that it is mandatory to design more studies of confined and unrestricted built environments with bigger sample sizes, more replicates and even deeper sequencing. Such attempts should help to determine broader validity of our results and in parallel benchmark bioinformatics tools to determine their capability to determine the resistome in dependence on microbial diversity, obtained biomass, and sequencing depth. Without further data we are stranded on the level of speculations. We included a paragraph about the limitations of our study in the discussion section.

“Beside this comprehensive analysis the present study faces some limitations, such as the low sample size from CB environments, its focus on one sample type (floor samples), and the lack of metadata on specific administered antibiotics especially in the ICU at the time of sampling in contrast to other studies ^{3,4}. This low sample size was a consequence of the restricted access to the confined built environment setting of the ICU and the cleanroom facility as well as the low amount of biomass in this CB environments. Hence, the representativeness of the subsequent analysis is limited and also constrained our attempts to correlate and interpret microbial, virulence or resistance compositions with environmental variables as it was shown in the study of Lax et al., in 2017 ³. Therefore the general validity and impact of the presented results require additional confirmation by further studies.”

Ln 441 “mandatory for cleanrooms to be almost void” is there a metric of ‘voidness’ – how is it mandatory?

A: You are right, there is no metric of ‘voidness’. Here we refer to the different ISO categories used to classify cleanrooms and determine cleanliness. We revised this sentence as follows:

“However, while it is mandatory for cleanrooms to be almost free of microorganisms...”

One thing that feels missing is an appropriate analysis of the genomic context – synteny – of the resistance genes, i.e. are ARGs co-associated with other resistance-genes, e.g. those to moisture deprivation, or nutrient acquisition. In Lax et al 2017 Sci Trans Med they suggested that the genomes of microbes that were enriched in patient rooms were enriched in co-associated genes encoding ARGs and host-invasion genes. So what’s going on in these organisms.

A: We added a comprehensive analysis of the resistance genes in their genomic context (co-localization with other genes, integron clusters, flanking repeats and potentially horizontally transferred genes including mobile genes and tRNA hotspots). We could also determine an enriched co-association with other ARGs, but not host-invasion genes for CB environments. Below, you can find the summary of this analysis:

“Resistance genes were further investigated in their genomic context (synteny). In most cases antibiotic resistance genes were co-localized with other resistance genes especially on genomes retrieved from CB environments (mainly multidrug efflux transporter systems e.g. *acrA*, *acrB* and *bepE*). In contrast, genomes from UB environments showed more often transcriptional regulators (e.g. *cymR* and *grpE*) and transposases (*tnpABC*) in close vicinity to annotated resistance genes. Despite the high frequency of transposase genes in vicinity to resistance genes, no integron clusters could be detected. Resistance genes of genomes from CB environments were also significantly more often surrounded by a higher frequency of flanking repeats ($P = 0.02$). Potentially horizontally transferred genes (HGT) in regions of genome plasticity (RGP) were identified by synteny breaks and compositional bias between genomes of CB and UB and closely related genomes available in the MaGe database 20. More potential HGT features (both mobility genes as well as tRNA hotspots) were detected in genomes from CB environments. However, higher proportions of HGT in CB were not significant.”

REVIEWERS' COMMENTS:

Reviewer #2 (Remarks to the Author):

I appreciate the authors' attempts to address my prior comments; however, I believe that the number of samples and approach taken does not allow for the quantitative comparisons being made. Simply put, the study is observational and does not allow for quantitative comparisons between environments. Specifically, the authors attempt to address my prior concerns with a 'limitations' discussion on L478-488, but this discussion seems to contradict the overall study presentation, e.g. L81-85 in the introduction. In addition, I believe much more careful wording is necessary around human health impacts, as this study does not actually demonstrate any health impact.

Reviewer #3 (Remarks to the Author):

The authors have done a lot of extra work and while they cannot fix the underlying flaws the analyses they have performed at least ameliorate the potential for false conclusions. I am happy to see this paper published.

Jack Gilbert

Response to Referees / Reviewers

Dear Reviewers,

thank you once again for your valuable comments! In the light of your suggestions we made some final edits to our manuscript.

We hope that you will appreciate them!

Sincerely yours

Alexander & coauthors

Reviewers' comments and Author replies (A:)

Reviewer #2 (Remarks to the Author):

I appreciate the authors' attempts to address my prior comments; however, I believe that the number of samples and approach taken does not allow for the quantitative comparisons being made. Simply put, the study is observational and does not allow for quantitative comparisons between environments. Specifically, the authors attempt to address my prior concerns with a 'limitations' discussion on L478-488, but this discussion seems to contradict the overall study presentation, e.g. L81-85 in the introduction. In addition, I believe much more careful wording is necessary around human health impacts, as this study does not actually demonstrate any health impact.

A: We revised the study presentation in the introduction and do not think that it still contradicts our discussion on the limitations of the present study.

“These new insights are useful to model human driven processes affecting in-house microbiota and its associated resistome and to improve our assessments on the possibilities of preserving or, eventually, designing microbiomes in built environments.”

Further, we had a detailed look on our wording around human health impacts. In this final revised version of our manuscript we never link our results to any health impacts. Only in the discussion section we cite the potential connection between microbial exposure and potent immune development and cover it by four references.

“However, an unselective removal and killing of many microbes in the built environment could have adverse health effects, since potent immune development may rely on microbial exposure^{23,28–30}.”

Therefore, we are convinced that the present study has its value and that our quantitative comparisons are meaningful despite the small sample size as we discuss all limitations in great detail.

Reviewer #3 (Remarks to the Author):

The authors have done a lot of extra work and while they cannot fix the underlying flaws the analyses they have performed at least ameliorate the potential for false conclusions. I am happy to see this paper published.

Jack Gilbert

A: Thank you so much for your constructive criticism and numerous suggestions to improve our manuscript. Due to your help, our manuscript definitely advanced and we are also happy to see this paper published!